# Estimation-uncertainty affects decisions with and without learning opportunities

Kristoffer C. Aberg ●[1,2] ✉, Levi Antle[1,3] & Rony Paz ●[1,2] ✉

Motivated behavior during reinforcement learning is determined by outcome expectations and their estimation-uncertainty (how frequently an option has been sampled), with the latter modulating exploration rates. However, although differences in sampling-rates are inherent to most types of reinforcement learning paradigms that confront highly rewarded options with less rewarded ones, it is unclear whether and how estimation-uncertainty lingers to affect long-term decisions without opportunities to learn or to explore. Here, we show that sampling-rates acquired during a reinforcement learning phase (with feedback) correlate with decision biases in a subsequent test phase (without feedback), independently from outcome expectations. Further, computational model-fits to behavior are improved by estimation-uncertainty, and specifically so for options with smaller sampling-rates/larger estimation-uncertainties. These results are replicated in two additional independent datasets. Our findings highlight that estimation-uncertainty is an important factor to consider when trying to understand human decision making.

Reinforcement learning refers to the ability to associate events, stimuli, or actions with rewards and punishments[1], and therefore plays an integral part in acquiring new skills, ranging from basic perceptual[2] and motor skills[3], to the development of higher-level cognitive strategies[4,5]. The identification of factors that contribute to reinforcement learning and decision making is therefore key in understanding normal - as well as abnormal - behavioral adaption.

An intrinsic property of many common reinforcement learning paradigms is that more rewarded options (i.e., Good options) are sampled much more frequently than less rewarded or punished options (i.e., Bad options). The purpose of sampling is to reduce an option's uncertainty (i.e., estimation-uncertainty), which increases the reliability of estimates of relevant behavioral factors (e.g., expected outcomes or expected values). Accordingly, because the estimation-uncertainty for frequently sampled Good options is small, expected value estimates are useful for guiding decision making. By contrast, other decision heuristics may be more appropriate for less sampled Bad options because their estimation-uncertainty is large.

Some evidence suggests that estimation-uncertainty itself has utility for decision making. For example, options that were scarcely sampled in a passive viewing phase were more likely to be selected in a subsequent phase with informative feedback[6,7], and options with more variable (vs. constant) outcomes were more frequently selected during learning[8,9]. Additionally, the time since an option was last sampled, a factor which in dynamic environments is monotonically and positively correlated with estimation-uncertainty, was used to guide decisions in a restless bandit-task, and individuals more sensitive thereof displayed increased exploration rates[10]. The utility of estimation-uncertainty likely depends on how much it can be reduced by informative feedback during learning (i.e., its prospective information-gain)[11], therefore suggesting a role for estimation-uncertainty *during* learning by incentivizing exploration.

However, it remains unclear if estimation-uncertainty affects decisions for the longer-term, namely when information-gains are not provided and learning hindered. On one hand, because estimation-uncertainty is believed to afford information-gains, it might only affect behavior during learning with informative feedback. On the other hand, other types of uncertainty - such as risk and ambiguity - are perceived as strongly aversive under no-learning conditions[12,13], with ambiguity being closely related to estimation-uncertainty[14].

The present study was designed to test two hypotheses. First, whether estimation-uncertainty acquired during learning lingers and

[1]Department of Brain Sciences, Weizmann Institute of Science, Rehovot, Israel. [2]Azrieli Institute for Brain and Neural sciences, Weizmann Institute of Science, Rehovot, Israel. [3]Department of Psychology, University of Toronto, Toronto, ON, Canada. ✉e-mail: kc.aberg@gmail.com; rony.paz@weizmann.ac.il

affects decisions in the absence of information-gains and further learning. Second, whether its impact is asymmetric with regards to the valence of the considered options (i.e., Good and Bad options). We designed a probabilistic reinforcement learning task with five different conditions, each consisting of pairs of one Good and one Bad option (two-armed bandit). The absolute expected values for Good and Bad options differed between conditions, but the difference between Good and Bad options within a condition was identical across conditions. In a subsequent test phase, participants ($n = 50$) made decisions between inter-mixed options from the learning phase using the learned information and without receiving feedback. Behavioral modeling was used to disentangle the contribution of different factors (e.g., estimation-uncertainty, expected value) during the test phase.

The results support the hypothesis that participants consider estimation-uncertainty both during learning and in the subsequent test phase without learning. In particular, model-agnostic evidence show that decisions involving only uncertain Bad options are less influenced by expected values (as compared to decisions with only Good options), as well as significant correlations between an options sampling-rate during learning and its selection-rate during the test phase. Critically, these effects cannot be explained by differences in expected values. Behavioral modeling supports these results by showing that estimation-uncertainty contributed significantly to test-phase decisions involving Bad options, but not to decisions between two Good options. These results were replicated in two additional experiments (100 participants each, publicly available datasets)[15].

Our study highlights how acquired estimation-uncertainty affects decisions without further learning. Failing to account for estimation-uncertainty limits our understanding of the motivational factors that drive behavior, including reinforcement learning biases related to psychopathology.

## Results

Participants performed a probabilistic reinforcement learning task with five different conditions, each consisting of pairs of one Good and one Bad option (probability of receiving positive/negative feedback being respectively 0.75/0.25 and 0.25/0.75; Fig. 1A, B). The conditions differed in the probability ($p_{Appetitive}$) that a feedback would be drawn from an appetitive context (positive/negative feedback = +1₪/0₪), or an aversive context (positive/negative feedback=0₪/-1₪). In other words, each option's expected value correlated positively with the value of $p_{Appetitive}$ (Supplementary Table 1). In a subsequent test phase, participants made decisions between pairs consisting of inter-mixed options from the learning phase (i.e., Good vs Bad, Good vs Good and Bad vs Bad pairs; Fig. 1C). During this phase, no feedback was provided. Besides behavioral measures, we use behavioral modeling to disentangle the contribution of different behavioral factors to performance (e.g., estimation-uncertainty, expected value), and replicate our initial findings by analyzing two separate independent experiments (100 participants each)[15].

### Estimation-uncertainty plays a role during learning

Participants learned the task in all five conditions (Fig. 1D). Across conditions, overall learning performance was higher than the chance-level [ANOVA intercept term vs. 0.5: $F(1,49) = 9.14$ $p < 0.001$, repeated measures ANOVA with one factor Condition], as well as separately in all conditions [Fig. 1E; all $p < 0.001$; Supplementary Table 2]. There was also an effect of Condition [$F(4196) = 4.57$, $p = 0.002$, $\eta_p^2 = 0.085$], with performance in the most rewarding condition ($p_{Appetitive}$=1.0) being higher than in all other conditions, and significantly so in two comparisons (Supplementary Table 3).

The model-recovery procedure confirmed that the tested models predicted different behavioral patterns (Fig. 1F; procedure described in Methods: Model recovery). Next, we obtained protected exceedance probabilities and model frequencies for the different models via

the HBI method (described in Methods: Model fitting)[16], with the Kalman:QU model providing the most parsimonious fit to behavior (Protected exceedance probability = 0.806; Fig. 1G; Pseudo-$R^2 = 0.293$). The Kalman:QU model contains two decision weights: $\beta_Q$ for expected values and $\beta_U$ for estimation-uncertainty (Fig. 1H), and both parameters were successfully recovered when used to generate simulated data (Fig. 1I; described in Methods: Parameter recovery). Beyond providing a better relative fit to behavior as compared to other models, there is a close similarity between model-fitted and actual performance, both in terms of learning curves (cf. Fig. 1D versus Fig. 1J) and average performance (cf. Fig. 1E versus Fig. 1K). The model also captures inter-individual differences in learning performance, as illustrated by significant and positive correlations between actual and model-fitted learning performances (all Pearson's r > 0.7; Fig. 1L, left panel). To illustrate the contribution of estimation-uncertainty, we compared these correlations with the corresponding correlations of the Kalman:Q model, which fits behavior based on expected values only (Fig. 1L, right panel). Permutation tests showed significantly stronger correlations for the Kalman:QU model in three out of five conditions (see Table 1). In summary, the Kalman:QU model, which contains both estimation-uncertainty and expected values, provides the most parsimonious fit to behavior during reinforcement learning.

### Estimation-uncertainty plays a role after learning

As a confirmation for successful transfer of learned information to the test phase, we first tested whether participants were more likely to select Good options in Good versus Bad pairs. These pairs were created by combining Good options from each condition with Bad options from the same condition, but from different learning blocks. The proportion of Good (vs Bad) selections was significantly higher than chance across all conditions [Fig. 2A; ANOVA intercept term vs. 0.5: $F(1, 49) = 103.3$, $p < 0.001$, repeated measures ANOVA with factor Condition], with no difference between conditions [Condition: $F(4, 196) = 1.44$, $p = 0.223$, $\eta_p^2 = 0.029$]. Hence, participants were able to transfer information acquired during the learning phase to the subsequent test phase.

Next, we tested for biases in the selections of options in Good versus Good and Bad versus Bad pairs. These pairs were created by combining options from different conditions within the same learning block (e.g., the Good options from $p_{Appetitive} = 1.0$ and $p_{Appetitive} = 0.5$, both learned in block 1). Both Good versus Good and Bad versus Bad comparisons show main effects of Condition [Fig. 2B; Good versus Good: $F(4196) = 12.64$, $p < 0.001$, $\eta_p^2 = 0.205$]; Fig. 2C; Bad versus Bad: $F(4196) = 6.021$, $p < 0.001$, $\eta_p^2 = 0.109$; Supplementary Table 4, repeated measures ANOVAs with factor Condition].

To better understand these results, the selection bias was analyzed in a pair-wise manner (Fig. 2D). For Good versus Good pairs, there is a clear selection bias favoring options from more rewarded conditions (i.e., with larger $p_{Appetitive}$; Fig. 2D, blue line). A similar, but weaker trend can be observed for Bad versus Bad pairs (Fig. 2D, pink line). Indeed, overall participants selected options that had been more frequently rewarded during the learning phase [ANOVA intercept term vs. 0.5: $F(1, 49) = 52.14$, $p < 0.001$, repeated measures ANOVA with factors Comparison and GG/BB; Supplementary Table 6], and were more able to do so in Good versus Good pairs, as compared to Bad versus Bad pairs [main effect of GG/BB [$F(1, 49) = 4.36$, $p = 0.042$, $\eta_p^2 = 0.082$; mean selection rates of the more rewarding option in GG/BB was 0.603/0.558]. This latter finding is well in accordance with the prediction that decision strategies for frequently selected Good options are more biased towards the use of expected-values, as compared to less frequently Bad options. Further, not surprisingly given large differences in expected values between options in different pairs, performance in some pairs was higher than in others [main effect of Comparison ($F(9441) = 3.37$, $p < 0.001$, $\eta_p^2 = 0.064$]. There was no interaction between GG/BB and Comparison [$F(9441) = 1.31$, $p = 0.232$,

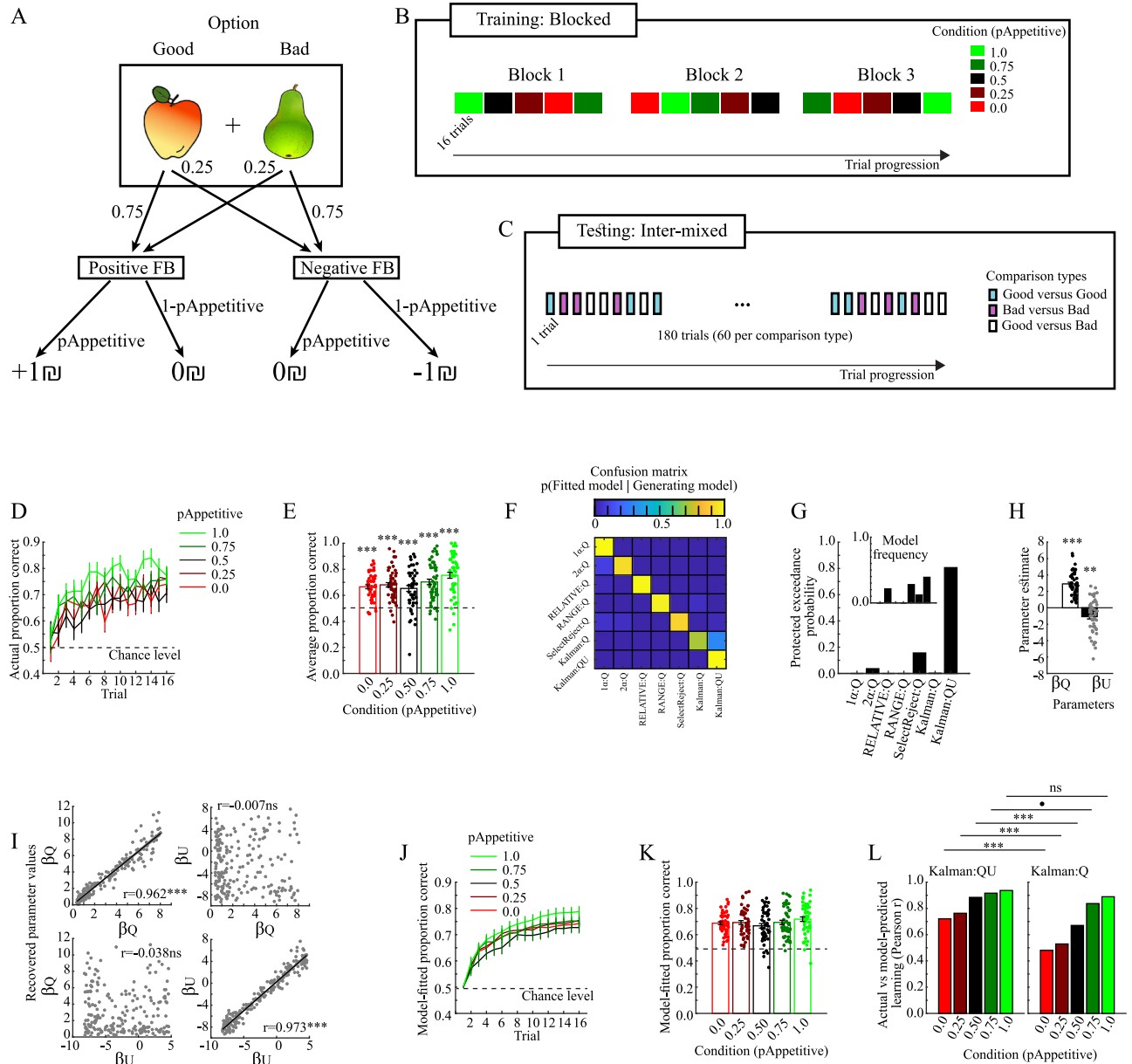

**Fig. 1 | Learning phase (*n* = 50 samples, unless stated otherwise). A** Schematic of stimulus-outcome contingencies. Stimulus images from the Snoddgrass and Vanderwart 'Like' Objects dataset[39] are released under a Creative Commons Attribution-NonCommercial-ShareAlike 3.0 Unported License by courtesy of Michel J. Tarr, Carnegie Mellon University http://tarrlab.org. **B** Schematic of trial progression during the learning phase. New objects were presented in each block, for a total of 15 different pairs of objects. **C** Schematic of trial progression during the test phase. **D** Actual learning curves. **E** Average actual learning performances ($t(49)_{1.0} = 10.886$, $p < 0.001$, 95%CI = [0.707 0.801], $d = 1.537$; $t(49)_{0.75} = 9.422$, $p < 0.001$, 95%CI = [0.660 0.746], $d = 1.500$; $t(49)_{0.5} = 6.689$, $p < 0.001$, 95%CI = [0.607 0.699], $d = 0.946$; $t(49)_{0.25} = 10.578$, $p < 0.001$, 95%CI = [0.646 0.715], $d = 1.333$; $t(49)_{0.0} = 10.870$, $p < 0.001$, 95%CI = [0.635 0.696], $d = 1.540$). **F** The confusion matrix obtained from the model-recovery procedure ($n = 200$ virtual participants). Denotation of a model's learning part is separated from its decision part by the ':' symbol, e.g., the Kalman:QU model learns using a Kalman-filter (Kalman) and decides based on expected values (Q) and estimation-uncertainties (U). **G** Protected exceedance probabilities and model-frequencies (inset) obtained from the model-fitting procedure. **H** Fitted values of the model-parameters for the winning Kalman:QU model ($t(49)_{βQ} = 14.399$, $p < 0.001$, 95%CI = [2.510 3.324],

Cohen's $d = 2.036$; $t(49)_{βU} = 3.790$, $p < 0.001$, 95%CI = [−1.653 −0.507], Cohen's $d = 0.536$). **I** Pearson's correlation coefficients between the values of generating model-parameters and their recovered values obtained via the parameter-recovery procedure ($n = 200$ virtual participants; $r_{βQ-βQ} = 0.962$, $p < 0.001$; $r_{βU-βU} = 0.973$, $p < 0.001$; $r_{βQ-βU} = 0.007$, $p = 0.919$; $r_{βU-βU} = −0.038$, $p = 0.598$). **J** Model-fitted learning curves. K. Average model-derived learning performances. **L** Pearson's correlation coefficients between actual and model-fitted performances in each condition for the Kalman:QU model (left panel) and the Kalman:Q model (right panel). $Δr_0 = 0.240$, $p_{Permutation} < 0.001$, 95%CI$_{Permutation}$ [−0.082 0.080], SES = 2.859; $Δr_{0.25}.25 = 0.234$, $p_{Permutation} = 0.003$, 95%CI$_{Permutation}$ [−0.092 0.093], SES = 3.083; $Δr_{0.50} = 0.214$, $p_{Permutation} = 0.002$, 95%CI$_{Permutation}$ [−0.139 0.135], SES = 3.004; $Δr_{0.75} = 0.079$, $p_{Permutation} = 0.083$, 95%CI$_{Permutation}$ [−0.157 0.144], SES = 1.755; $Δr_{1.0} = 0.047$, $p_{Permutation} = 0.276$, 95%CI$_{Permutation}$ [−0.174 0.155], SES = 1.092. All tests are two-tailed and uncorrected $p$ values are reported. All errorbars indicate the standard error of the mean, and the shaded areas of Figure I indicate 95% confidence interval. d=Cohen's d. SES=Standardized effect size. $p_{Permutation}$ and 95% CI$_{Permutation}$ is the p-value and confidence interval for the null-distribution obtained via permutation testing. ***$p < 0.001$, **$p < 0.01$, *$p < 0.05$, •$p < 0.10$, ns not significant.

**Table 1 | Difference in correlation coefficients between actual and model-fitted learning performance for the Kalman:QU versus the Kalman:Q model in Experiment 1**

| Condition ($p_{Appetitive}$) | r Kalman:QU | r Kalman:Q | Δ r | p-value | 95% confidence interval of permutation distribution | Standardized effect size (SES) |
|---|---|---|---|---|---|---|
| 1.0 | 0.937 | 0.890 | 0.047 | 0.276 | −0.174 0.155 | 1.092 |
| 0.75 | 0.916 | 0.837 | 0.079 | 0.083 | −0.157 0.144 | 1.755 |
| 0.5 | 0.884 | 0.670 | 0.214 | 0.002 | −0.139 0.135 | 3.004 |
| 0.25 | 0.763 | 0.529 | 0.234 | 0.003 | −0.092 0.093 | 3.083 |
| 0.0 | 0.721 | 0.481 | 0.240 | <0.001 | −0.082 0.080 | 2.859 |

r is Pearson's correlation coefficient between actual and model-fitted learning performances. Uncorrected two-tailed p-values, confidence intervals, and standardized effects sizes were obtained from permutation tests ($n = 1000$ permutations).

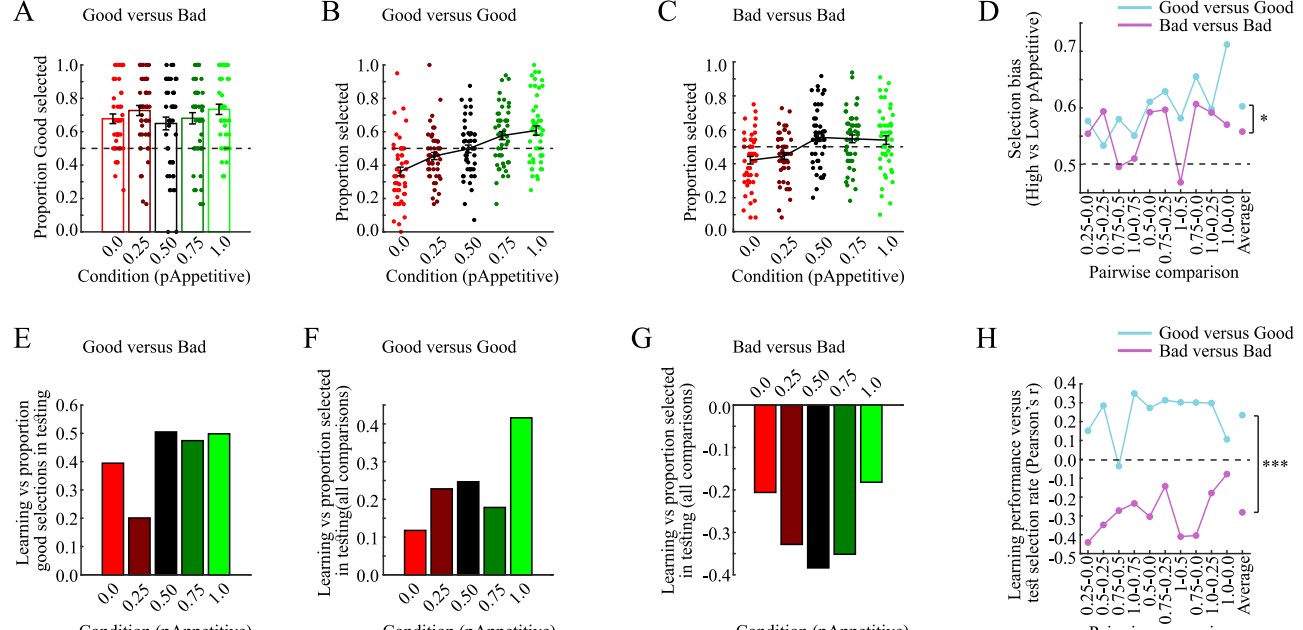

**Fig. 2 | Test phase behavior ($n = 50$, unless stated otherwise). A** Above-chance selection rates of the Good option in Good versus Bad pairs indicate that participants successfully transferred learned information to the test phase. These comparisons contrast Good and Bad options from one condition learned in different blocks. **B** Selection rates of options from different conditions in Good versus Good pairs. These comparisons contrast Good options from different conditions, learned in the same block. **C** Selection rates of options from different conditions in Bad versus Bad pairs. These comparisons contrast Bad options from different conditions, learned in the same block. **D** Selection bias towards more valuable options in pairwise comparisons. These results indicate that options with higher expected values in the learning phase were more likely to be selected in the test phase, in particular for Good versus Good pairs (ANOVA, main effect: F(1, 49) = 4.36, $p = 0.042$, $\eta_p^2 = 0.082$). **E** Pearson's correlation coefficients between learning performance and the selection rate of Good options in Good versus Bad pairs. Participants with higher learning performance were more likely to select the Good

option. **F** Pearson's correlation coefficients between learning performance and the selection of Good options in Good versus Good pairs. Participants with higher learning performance in a condition were more likely to select Good options from that condition. **G** Pearson's correlation coefficients between learning performance and the selection of Bad options in Bad versus Bad pairs. Participants with higher learning performance in a condition were less likely to select Bad options from that condition. **H** Pairwise correlations (Pearson) between the difference in learning performance and selection rates in the test phase. Differences in learning performance correlated positively with differential selection rates in Good vs Good pairs (blue line), but negatively with the difference in selection rates in Bad vs Bad pairs (pink line), with a significant difference between these two pair types (ANOVA, main effect: F(1, 2839) = 112.4, $p < 0.001$). Please observe that the learning performance is exactly related to the sampling-rate of Good options, but inversely so (1-learning performance) for Bad options. $\eta_p^2$=Partial eta squared. ***$p < 0.001$, *$p < 0.05$.

$\eta_p^2 = 0.026$]. Next, we directly compared selection biases for Good versus Good and Bad versus Bad pairs with the difference in expected values and their entropy (Supplementary Table 1). As would be predicted, we observed a significant correlation for expected values in Good versus Good pairs [Pearson's r = 0.879, $p < 0.001$; Spearman's ρ = 0.890, $p < 0.001$] but not for Bad versus Bad pairs [Pearson's r = 0.396, $p = 0.180$; Spearman's ρ = 0.356, $p = 0.313$]. By contrast, selections were positively biased towards higher-entropy options in Bad versus Bad pairs [Pearson's r = 0.636, $p = 0.048$; Spearman's ρ = 0.697, $p = 0.031$], but not for Good versus Good pairs [Pearson's r = 0.238, $p = 0.509$; Spearman's ρ = 0.152, $p = 0.682$]. Because both

expected values and entropy may contribute to an option's sampling rate during learning, next we test for direct connections between sampling rates during learning and selection biases during testing.

**Estimation-uncertainty during learning affects decisions after learning**

In a static environment, estimation-uncertainty is monotonically and negatively correlated with an options sampling-rate. Specifically, for a Good option the sampling-rate is exactly equal to the learning performance in the pair in which it was learned (i.e., how often the Good option was selected divided by the total number of trials), while for the

corresponding Bad option the sampling-rate is exactly equal to one minus that learning performance. Accordingly, a critical model-agnostic comparison is between an option's sampling-rate during learning and its selection-rate during testing. For this reason, we tested the link between the learning performance within a condition and the selection rate of the corresponding Good and Bad options during the test phase, using a linear mixed effects model which allows different values of the continuous covariate Learning performance for the different Conditions (five levels of $p_{Appetitive}$).

For Good versus Bad pairs, higher learning performance in a condition was positively associated with more frequent selections of Good options from that condition [main effect Learning performance: $F(1244) = 11.86$, $p < 0.001$], and was not affected by the condition itself [Interaction between Condition and Learning performance: $F(1244) = 1.41$, $p = 0.232$]. In sum, for all conditions, test selections of Good options were positively correlated with their corresponding learning performance (Fig. 2E).

Similar results were obtained for Good versus Good pairs. Higher learning performance in a condition was positively associated with more frequent selections of the Good option from that condition in the test phase [main effect Learning performance: $F(1244) = 11.86$, $p < 0.001$], and was not affected by the condition itself [Interaction between Condition and Learning performance: $F(1244) = 0.75$, $p = 0.557$]. In sum, for all conditions, selections of Good options were positively correlated with learning performances (Fig. 2F).

Similar, but reversed, results were obtained for Bad versus Bad pairs. Higher learning performance in a condition was negatively associated with the selection of Bad options from that condition [main effect Learning performance: $F(1244) = 19.53$, $p < 0.001$], and was not affected by the condition itself [Interaction between Condition and Learning performance: $F(1244) = 0.40$, $p = 0.811$]. In sum, for all conditions, selections of Bad options during the test phase were negatively correlated with learning performance (Fig. 2G).

In addition, there was a significant interaction between learning performance and relative proportion of selections in the test phase. Namely, we correlated the difference in learning performance between two conditions, with the relative proportion of selections in Good versus Good and Bad versus Bad pairs [$F(1, 2839) = 112.4$, $p < 0.001$, linear mixed-effects modeling with factors GG/BB, Comparison (10 different pairwise comparisons), and Block (3 learning blocks); all other main factors $p > 0.05$; Table 2]. The interaction was caused by respectively positive and negative correlations between differences in Learning performance for Good versus Good [average Pearson's $r = 0.234$] and Bad versus Bad comparisons [average Pearson's $r = -0.281$] (Fig. 2H, Individual correlation coefficients). These results have two important implications. First, the negative correlation between learning performance and the selection bias for Bad options with higher expected values suggests that learning performance does not provide a straightforward estimate of what information has been learned. Instead, the selection bias of Bad options was more closely associated with how frequently it had been sampled during the learning phase (i.e., the sampling-rate for Bad options in a condition is precisely 1-learning learning performance). Of note, the lack of interactions with factor Block indicates that more recently learned options did not show differences in the susceptibility of estimation-uncertainty effects, something which may have been the case given relevant interactions between reinforcement learning and working memory[17,18].

In summary, this model-agnostic approach shows that participants were consistently more likely to select Good options from conditions where previous learning performance was relatively higher, whereas Bad options were less likely to be selected when drawn from contexts with higher learning performance. Importantly, this latter finding cannot be explained by differences in expected values, because Bad options from high reward conditions (i.e., with relatively larger values of $p_{Appetitive}$) were more likely to be rejected. Instead, and in accordance with our hypothesis, participants are likely to avoid Bad options with large estimation-uncertainties, which are monotonically and negatively proportional to their sampling-rates during learning. To further support this interpretation, we applied behavioral modeling to disentangle the contribution of expected values and estimation-uncertainty to behavior during the test phase.

The model-recovery procedure shows that all tested models provide distinguishable behaviors in the test phase (Fig. 3A). Further, model-comparisons show that the Kalman:QU model provides the most parsimonious fit to behavior (Fig. 3B; protected exceedance probability = 0.602; Pseudo-$R^2 = 0.169$), followed by the RANGE:Q model (protected exceedance probability = 0.398). For the Kalman:QU model, fitted model-parameters were successfully recovered (Fig. 3C, D), and the modeled selection rates for Good versus Good and Bad versus Bad pairs closely match actual behavior (Fig. 3E, cf. Fig. 2B, C). Next we tested to what extent the different models capture inter-individual biases in behavior via linear mixed-effects ANOVAs with factors Condition and GG/BB, with actual selection bias as dependent variable, and the model-fitted selection bias as continuous covariate. For all the three models, the model-fitted selection bias was significant [Table 3; Kalman:QU: $F(1979) = 258.51$, $p < 0.001$; Kalman:Q: $F(1979) = 105.4$, $p < 0.001$; RANGE:Q: $F(1979) = 74.16$, $p < 0.001$], indicating strong correlations between actual and model-fitted selection rates (Fig. 3F–H). However, the RANGE:Q and Kalman:Q models also displayed significant interactions between the model-fitted selection bias and GG/BB [Kalman:QU: $F(1979) = 0.12$, $p = 0.728$; Kalman:Q: $F(1979) = 15.22$, $p < 0.001$; RANGE:Q: $F(1979) = 105.4$, $p < 0.001$]. As seen in Fig. 3F, the Kalman:QU model provides equally good fits for Good versus Good and Bad versus Bad pairs, while the two other models show worse fits for Bad versus Bad pairs (Fig. 3G, H). Indeed, directly contrasting the performance of the Kalman:QU model with the other two models confirmed that the Kalman:QU model provided a significantly better fit to behavior in Bad versus Bad, as compared to Good versus Good pairs [versus RANGE:Q model; Fig. 3I; mean difference in Pearson's $r = 0.173$, $t(9) = 5.197$, $p < 0.001$, 95%CI = [0.098 0.249], Cohen's $d = 1.643$; versus Kalman:Q model; Fig. 3J; mean difference in Pearson's $r = 0.214$, $t(9) = 5.813$, $p < 0.001$, 95%CI = [0.131 0.297], Cohen's $d = 1.838$].

In summary, the Kalman:QU model provides the best fit to behavior during the test phase. Importantly, the impact of estimation-uncertainty was significant for Bad versus Bad pairs, but not for Good versus Good pairs, indicating a dissociation between decision heuristics used for reliable (i.e., Good versus Good) and unreliable (i.e., Bad versus Bad) options.

## Replication of results in two independent datasets

To replicate our findings, we analyzed two additional behavioral datasets. Because behavior has been described extensively elsewhere (published by Bavard, Rustichini, Palminteri[15]), we show here only the relevant comparisons with the corresponding behaviors from the model we developed here.

In the first dataset, pairs of objects were learned and tested in an inter-mixed fashion (Fig. 4A–C). Learning and test performances are shown in Fig. 4D, E. Replicating our previous model-agnostic results, there were positive correlations between learning performance and selection biases in the test phase. Specifically, for Good versus Good pairs, learning performance in A1B1 (vs. C1D1) pairs is positively correlated with the selection rate of A1 options in A1C1 pairs [Pearson's $r = 0.207$, $p = 0.039$; Spearman's $\rho = 0.229$, $p = 0.022$; Fig. 4F]. By contrast, for Bad versus Bad pairs, the learning performance is negatively correlated with the selection rate of B1 options in B1D1 pairs [Pearson's $r = -0.300$, $p = 0.003$; Spearman's $\rho = -0.272$, $p = 0.006$; Fig. 4G]. The Kalman:QU model provides the best fit to behavior [Fig. 4H; protected exceedance probability = 0.983; Pseudo-R2 = 0.235], and provides a close match to the actual behavior (Fig. 4I, cf. Fig. 4D). Fitted model

**Table 2 | Correlation between difference in learning performance and selection rates as a function of Condition (ten pairwise comparisons), GGvBB (i.e., Good versus Good, Bad versus Bad), and Block (i.e., three learning blocks) in Experiment 1**

| Predictor | df 1 | df 2 | F | p value |
|---|---|---|---|---|
| Learning performance | 1 | 2839 | 0.49 | 0.484 |
| Learning performance x Block | 2 | 2839 | 0.22 | 0.799 |
| Learning performance x Condition | 9 | 2839 | 0.82 | 0.600 |
| Learning performance x GGvBB | 1 | 2839 | 112.4 | <0.001 |
| Learning performance x GGvBB x Block | 2 | 2839 | 2.21 | 0.110 |
| Learning performance x GGvBB x Condition | 9 | 2839 | 0.68 | 0.728 |

P-values were obtained from ANOVA on the outputs of a linear-mixed effects model. F is the F-statistics, and df1 and df2 are degrees of freedom.

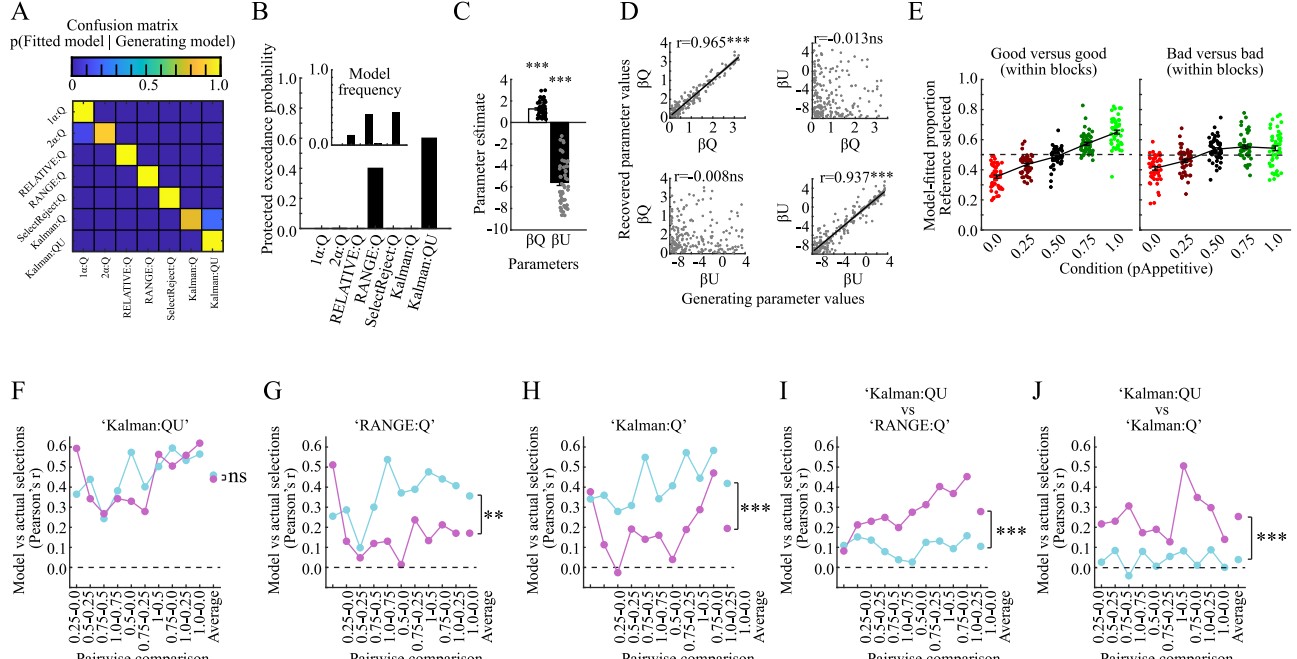

**Fig. 3 | Test phase modeling ($n = 50$, unless stated otherwise). A** Confusion matrix obtained from the model-recovery procedure. **B** Protected exceedance probabilities and model frequencies (inset). **C** Fitted values for the model-parameters of the Kalman:QU model (t(49)$_{\beta Q}$ = 11.979, $p < 0.001$, 95%CI = [1.020 1.430], d = 1.721; t(49)$_{\beta U}$ = 18.566, $p < 0.001$, 95%CI = [−6.083 4.896], d = 2.689). **D** Pearson's correlation coefficients between the values of generating model-parameters and their recovered values ($r_{\beta Q\cdot\beta Q}$ = 0.965, $p < 0.001$; $r_{\beta U\cdot\beta U}$ = 0.937, $p < 0.001$; $r_{\beta Q\cdot\beta U}$ = −0.013, $p = 0.855$; $r_{\beta U\cdot\beta U}$ = −0.008, $p = 0.909$). **E** Model-predicted selection rates of options from different conditions in Good versus Good (left panel) and Bad versus Bad pairs (right panel). **F** Correlations (Pearson) between actual and model-fitted performances in each pairwise comparison for the Kalman:QU model. There is no significant difference in correlations between Good versus Good (blue line) and Bad versus Bad pairs (pink line; ANOVA main effect F(1, 979) = 0.12, $p = 0.728$). **G** Correlations (Pearson) between actual and model-fitted performances in each pairwise comparison for the RANGE:Q model show a difference in correlations between Good versus Good (blue line) and Bad versus Bad

pairs (pink line; ANOVA main effect F(1, 979) = 105.4, $p < 0.001$). **H** Correlations (Pearson) between actual and model-fitted performances in each pairwise comparison for the Kalman:Q model show a difference in correlations between Good versus Good (blue line) and Bad versus Bad pairs (pink line; ANOVA main effect F(1, 979) = 15.22, $p < 0.001$). **I** Differential correlations (Pearson) between actual and model-fitted performances in each pairwise comparison for the Kalman:QU versus the RANGE:Q model. The Kalman:QU model provides a better fit for Bad versus Bad pairs (pink line; $t(49)$ = 5.197, $p < 0.001$, 95%CI = [0.098 0.249], $d = 1.643$). **J** Differential correlations (Pearson) between actual and model-fitted performances in each pairwise comparison for the Kalman:QU versus the Kalman:Q model. The Kalman:QU model provides a better fit for Bad versus Bad pairs (pink line; $t(49)$ = 5.813, $p < 0.001$, 95%CI = [0.131 0.297], d = 1.838). All reported tests are two-tailed and uncorrected $p$ values are reported. All errorbars indicate the standard error of the mean, and the shaded areas of D indicate 95% confidence interval. All errorbars indicate the standard error of the mean. d=Cohen's d. ***$p < 0.001$, **$p < 0.01$, *$p < 0.05$, ns not significant.

parameters are shown in Fig. 4J. Importantly, as compared to the Kalman:Q model without estimation-uncertainty, the Kalman:QU model improves fits to behavior in all conditions, and particularly so in the conditions with low reward (Fig. 4K; Table 4; permutation tests). In brief, the Kalman:QU model provides the best fit to behavior during the learning in this task.

The Kalman:QU model provides the best fit to behavior also in the test phase (Fig. 4L; protected exceedance probability = 0.997; Pseudo-R2 = 0.195), and closely matches the actual behavior (Fig. 4M, cf. Fig. 4E). Fitted model parameters are shown in Fig. 4N. The

Kalman-model with estimation uncertainty, as compared to the Kalman:Q model, provides improved fits to behavior in all conditions, and significantly so in the conditions containing a Bad option [Fig. 4O; Table 5; permutation tests]. In summary, the Kalman:QU model provides the best fit to behavior during the test phase, and specifically provides a better fit for decisions containing a Bad options.

In the second dataset, pairs of objects were learned and tested in a block-wise fashion[15] (Fig. 5A–C). We repeated the same analyses as above, and obtained largely the same model-agnostic results, with learning performance showing respectively positive and negative

correlations with selection biases in Good versus Good and Bad versus Bad pairs [Good versus Good: Fig. 5F; Pearson's r = 0.201, p = 0.045; Spearman's ρ = 0.225, p = 0.025; Bad versus Bad: Fig. 5G; Pearson's r = −0.139, p = 0.167; Spearman's ρ = −0.243, p = 0.015]. For the learning phase, the Kalman:QU model provides the best fit to behavior (Fig. 5H; protected exceedance probability = 1.0; Pseudo-R2 = 0.100), and provides a close match to the actual behavior (Fig. 5I, cf. Fig. 5D). Fitted model-parameters are shown in Fig. 5J. Again, the Kalman:QU model (vs. the Kalman:Q model) improves model-fits in the conditions with low reward (Fig. 5K; Table 4). For the test phase, the Kalman:QU model provides the best fit to behavior (Fig. 5L−N; protected exceedance probability = 0.91; Pseudo-R2 = 0.195), and significantly improves fits to behavior in all comparisons containing a Bad option (Fig. 5O; Table 5).

overall, in our own behavioral experiment, as well as in two additional datasets, a model that includes estimation-uncertainty (Kalman:QU model) provides the best fit to behavior during the learning, but importantly also during the test phase that occurs after learning and without feedback. Estimation-uncertainty contributes mainly by improving the fit to decisions that contain a Bad option.

## Additional analyses

To strengthen the main results, some additional analyses were performed. While the results are briefly reviewed here, full details are provided in Supplementary Notes 1–4.

In a first analysis (Supplementary Note 1), we tested whether the decision weights for expected value and estimation-uncertainty differed across the training and the test phase, and whether these effects were consistent across the three experiments. One experiment (conducted in the lab) showed a stronger preference for expected values during the training phase, while the two other experiments (conducted online) showed no significant differences (Supplementary Fig. 1A). Furthermore, two of the experiments (in which pairs were presented in an inter-mixed fashion during testing) showed increased aversion to uncertainty during the test phase, while the third experiment (with block-wise test pair presentation) showed no difference (Supplementary Fig. 1B). In essence, there was no evidence of consistent differences in decision weights between the training and the test phases across the three experiments. However, because there were some indications that such differences may depend on experimental parameters (e.g., online versus in-lab testing, and/or mixed versus blocked presentation of stimulus pairs during testing), future studies may benefit from considering such parameters during the experimental design.

In a second analysis (Supplementary Note 2), we tested whether adding the estimation uncertainty component from the Kalman-filter approach also improved the model fits for the non-Kalman models. Indeed, in 29 out of 30 pairwise comparisons (five different models x three experiments x two phases), the estimation uncertainty component enhanced model performances, as compared to the same model without estimation uncertainty (Supplementary Fig. 1C). This analysis further strengthens the notion that estimation uncertainty needs to be considered when trying to understand human decision making in similar reinforcement learning tasks.

In a third analysis (Supplementary Note 3), we tested whether the Kalman:QU model was able to reproduce the small magnitude effect observed during the learning phase (i.e., better performance for high vs. low EV pairs), as reported by Bavard et al.[15]. Indeed, while the model reproduced the small magnitude effect for all comparisons in each experiment (Supplementary Table 6), we also observed significantly positive correlations between actual- and model-reproduced magnitude effects. This latter result demonstrates the ability of the model to reproduce the small magnitude effect on an individual level.

In a fourth analysis (Supplementary Note 4), we tested which models provided the best explanation of behavior in the six excluded

**Table 3 | Correlation between actual and model-fitted selection bias as a function of Condition (ten pairwise comparisons) and GvG BvB (i.e., Good versus Good, Bad versus Bad) in Experiment 1 for different models**

| Predictor | df 1 | df 2 | F | p value |
|---|---|---|---|---|
| Kalman:QU model | | | | |
| Model-fitted bias | 1 | 979 | 258.51 | <0.001 |
| Model-fitted bias x GvG BvB | 1 | 979 | 0.12 | 0.728 |
| Model-fitted bias x Condition | 9 | 979 | 1.55 | 0.127 |
| Model-fitted bias x GvG BvB x Condition | 9 | 979 | 0.62 | 0.780 |
| RANGE:Q model | | | | |
| Model-fitted bias | 1 | 979 | 74.16 | <0.001 |
| Model-fitted bias x GvG BvB | 1 | 979 | 10.08 | 0.002 |
| Model-fitted bias x Condition | 9 | 979 | 0.93 | 0.495 |
| Model-fitted bias x GvG BvB x Condition | 9 | 979 | 1.12 | 0.343 |
| Kalman:Q model | | | | |
| Model-fitted bias | 1 | 979 | 105.4 | <0.001 |
| Model-fitted bias x GvG BvB | 1 | 979 | 15.22 | <0.001 |
| Model-fitted bias x Condition | 9 | 979 | 1.32 | 0.224 |
| Model-fitted bias x GvG BvB x Condition | 9 | 979 | 0.82 | 0.597 |

Uncorrected p-values were obtained from ANOVA on the outputs of a linear-mixed effects model. F is the F-statistics, and df1 and df2 are degrees of freedom.

experiments provided by Bavard et al. [15]. As mentioned previously, these six experiments were not included in the main study because they cannot be used to address our main hypotheses of whether and how sampling-rates and uncertainty acquired during learning with partial feedback affect subsequent decision biases (see Methods for further details). In brief, this analysis showed better performance for the Kalman:QU model during partial feedback conditions (Supplementary Fig. 2A–D), while range-adaptation models provided better fits during complete feedback conditions, and particularly so during the test phase (Supplementary Fig. 2E–L).

## Discussion

The present study aims to clarify whether estimation-uncertainty, a behavioral factor known to affect decision making *during* reinforcement learning, and which is monotonically and negatively related to an option's sampling-rate, can also affect and change decisions in a subsequent test phase without feedback. We show that, in three separate experiments, participants considered estimation-uncertainty during the learning phase, but importantly, also in the test phase. Test phase decisions were particularly affected for Bad options, and we demonstrate that it is attributed to their unreliability due to the low sampling-rate (model-agnostic results) / large estimation-uncertainty (modeling results).

Our study therefore reports an impact of acquired estimation-uncertainty on decisions even without potential information gains. As such, our results add to previous research showing how other types of uncertainty, such as risk, contributes to decision making[12,13,19,20]. Moreover, by showing a significant impact of naturally occurring differences in estimation-uncertainty during learning, we extend previous research using paradigms that experimentally control sampling-rates and estimation-uncertainty during learning. For example, a recent set of studies showed that options with variable outcomes, for

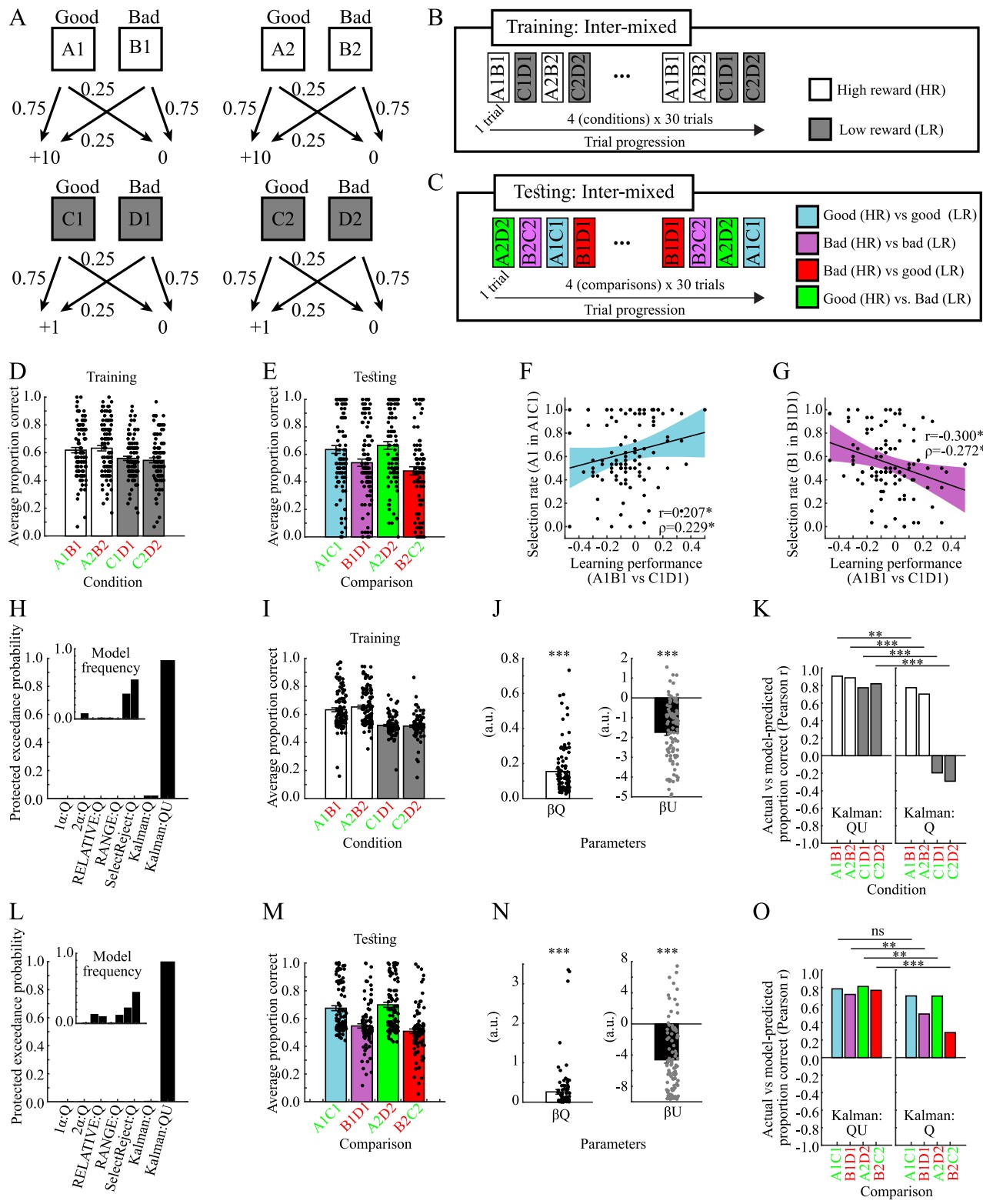

which feedback reduces their estimation-uncertainties, are preferred over options with constant outcomes (i.e., for which feedback provides no information gain)[8,9]. Further, options sampled one time during a passive viewing phase in the 'Horizon' task were more likely to be selected in subsequent conditions where feedback provided information which could be used to improve future decisions (horizon=6), as compared to conditions in which feedback provided no useful information gain (horizon=1)[6,7]. Finally, we recently observed that individuals with an increased sensitivity to estimation-uncertainty

were more likely to explore in dynamic environments where estimation-uncertainty is proportional to the time since an option was last sampled[10].

Our study highlights the need to consider factors beyond the experimentally controlled ones (e.g., expected values) when trying to understand motivated behavior in the context of reinforcement learning. As we discuss below, ignoring such factors may have consequences, ranging from the interpretation of behavior to the accuracy of behavioral models used to understand them.

**Fig. 4 | Validation 1 ($n = 100$). A** Schematic of stimulus-outcome contingencies **B** Schematic of trial progression during the learning phase. **C** Schematic of trial progression during the test phase. **D** Average actual learning performance. **E** Average actual test performance. **F** Positive correlation between differences in learning performance and selection biases in the test phase for Good versus Good pairs (r = 0.207, p = 0.039; ρ = 0.229, p = 0.022). **G** Negative correlation between differences in learning performance and selection biases in the test phase for Bad versus Bad pairs (r = −0.300, p = 0.003; ρ = −0.272, p = 0.006). Please observe that the learning performance is inversely related to the sampling-rate of Bad options (i.e., sampling-rate of Bad options = 1-learning performance). **H** Protected exceedance probabilities and model frequencies (inset) for the learning phase. **I** Average model-fitted learning performance. **J** Fitted values for the model-parameters of the Kalman:QU model (t(49)$_{\beta Q}$ = 10.938, p < 0.001, 95%CI = [0.126 0.182], d = 1.094; t(49)$_{\beta U}$ = 11.699, p < 0.001, 95%CI = [−2.072 −1.471], d = 1.170). **K** Correlations between actual and model-fitted learning performance in each condition for the Kalman:QU (left panel) and the Kalman:Q model (right panel). The Kalman:QU model provides a better fit in all conditions: Δr$_{A1B1}$ = 0.131, p$_{Permutation}$ = 0.002, 95% CI$_{Permutation}$ [−0.084 0.081], SES = 3.009; Δr$_{A2B2}$ = 0.186, p$_{Permutation}$ < 0.001, 95%

CI$_{Permutation}$ [−0.086 0.088], SES = 4.103; Δr$_{C1D1}$ = 0.972, p$_{Permutation}$ < 0.001, 95% CI$_{Permutation}$ [−0.310 0.329], SES = 5.987; Δr$_{C2D2}$ = 1.112, p$_{Permutation}$ < 0.001, 95% CI$_{Permutation}$ [-0.318 0.297], SES = 6.987. **L** Protected exceedance probabilities and model frequencies (inset) for the test phase. **M** Average model-fitted test performance. **N** Fitted values for the model-parameters of the Kalman:QU model (t(49)$_{\beta Q}$ = 4.402, p < 0.001, 95%CI = [0.141 0.371], d = 0.440; t(49)$_{\beta U}$ = 10.770, p < 0.001, 95%CI = [−5.561 −3.831], d = 1.077). **O** Correlations between actual and model-fitted performance in each comparison for the Kalman:QU (left panel) and the Kalman:Q model (right panel). The Kalman:QU model provides a better fit in all conditions including Bad options, but not the condition which includes two Good options (A1C1; blue bar; Δr$_{A1C1}$ = 0.081, p$_{Permutation}$ = 0.120, 95%CI$_{Permutation}$ [−0.088 0.087], SES = 1.338; Δr$_{B1D1}$ = 0.223, p$_{Permutation}$ = 0.002, 95%CI$_{Permutation}$ [−0.141 0.148], SES = 2.962; Δr$_{A2D2}$ = 0.110, p$_{Permutation}$ = 0.006, 95%CI$_{Permutation}$ [-0.078 0.078], SES = 2.762; Δr$_{B2C2}$ = 0.482, p$_{Permutation}$ < 0.001, 95%CI$_{Permutation}$ [−0.201 0.197], SES = 4.742). All errorbars indicate the standard error of the mean. d=Cohen's d. SES=Standardized effect size. p$_{Permutation}$ and 95%CI$_{Permutation}$ is the p-value and confidence interval for the null-distribution obtained via permutation testing. ***p < 0.001, **p < 0.01, *p < 0.05, ns not significant.

**Table 4 | Difference in correlation coefficients between actual and model-fitted learning performance for the Kalman:QU versus the Kalman:Q model in publicly available data sets 1 and 2**

| Condition | r Kalman:QU | r Kalman:Q | Δ r | p-value | 95% confidence interval of permutation distribution | Standardized effect size (SES) |
|---|---|---|---|---|---|---|
| Data set 1 | | | | | | |
| A1B1 | 0.908 | 0.776 | 0.131 | <0.001 | −0.084 0.081 | 3.009 |
| A2B2 | 0.889 | 0.703 | 0.186 | <0.001 | −0.086 0.088 | 4.103 |
| C1D1 | 0.776 | −0.196 | 0.972 | <0.001 | −0.310 0.329 | 5.987 |
| C2D2 | 0.821 | −0.291 | 1.112 | <0.001 | −0.318 0.297 | 6.987 |
| Data set 2 | | | | | | |
| A1B1 | 0.951 | 0.808 | 0.143 | 0.005 | −0.104 0.098 | 2.823 |
| A2B2 | 0.938 | 0.800 | 0.138 | 0.005 | −0.100 0.087 | 2.782 |
| C1D1 | 0.889 | −0.544 | 1.433 | <0.001 | −0.356 0.371 | 7.953 |
| C2D2 | 0.852 | −0.488 | 1.340 | <0.001 | −0.345 0.352 | 7.351 |

r is Pearson's correlation coefficient between actual and model-fitted learning performances. Two-tailed uncorrected p-values, confidence intervals, and standardized effects sizes were obtained from permutation tests ($n = 1000$ permutations).

To illustrate the former, we consider a frequently used reinforcement learning paradigm where participants first learn with three types of pairs (AB, CD, EF) that differ in the reward probability for the Good and Bad options (p$_{Reward}$ for A and B options=0.8 and 0.2, C/D = 0.7/0.3, and E/F = 0.6/0.4). In a subsequent test phase without feedback new pairs are presented, i.e., pairs of Bad versus Bad options (i.e., BD, BF, and DF) and Good versus Good options (i.e., AC, AE, and CF). An increased avoidance of B in the test phase is commonly interpreted as better punishment learning, while an increased selection of A indicates better reward learning[21–23]. However, our results propose a different interpretation. Specifically, the Bad B option in easy-to-learn AB pairs will be sampled less frequently than the Bad options in more difficult CD and EF pairs (i.e., C and E). As such, the B option is not only associated with a lower expected value, but also a larger estimation-uncertainty. Because the impact of these two factors cannot be disentangled on the behavioral level (both may lead to increased aversion), it is unclear whether the ability to avoid selecting the B is due to punishment learning or uncertainty avoidance. Our findings here suggest that in this and other similar tasks, the contribution of estimation-uncertainty requires further clarification, in order to allow an accurate understanding of the neurocomputational bases of observed reinforcement learning biases. This is important because the above mentioned task, and many other similar tasks, are frequently used to identify biases associated with various neurological conditions[23,24], genetic predispositions[25], as well as mental and behavioral disorders[26–29] and vulnerabilities thereof[22].

Moreover, our findings suggest that any Bad option, and hence more likely to be frequently avoided during learning, will suffer from a high estimation-uncertainty after the learning phase has ended (assuming that feedback is provided for selected options only). This estimation-uncertainty may contribute to subsequent avoidance decisions when the option is paired with more certain options, even when these are less valuable. Indeed, a number of studies report no clear selection bias when comparing Bad options learned in high reward conditions to Good options learned in low reward conditions[15,30], yet most studies do not test the contribution of estimation-uncertainty. We re-analyzed data from two such experiments, generously offered in an online repository[15], and show that models including estimation-uncertainty provide more parsimonious fits to behavior as compared to previously fitted models. In addition, a supplementary analysis showed that adding estimation uncertainty to the decision phase enhanced model performances across all tested models. These results confirm the need to consider estimation-uncertainty when modeling behavior in tasks that do not provide feedback for unselected options (i.e., partial feedback).

Of note, the present study was specifically designed to test whether and how naturally arising differences in estimation-uncertainty - due to learning with partial feedback - affect subsequent decision making. We therefore make no claims regarding behavioral biases expressed in complete feedback conditions where, presuming that all feedbacks are attended and/or processed in an unbiased manner, each option is sampled equally many times (e.g., Palminteri et al.[30]). However, because similar behavioral biases have been observed in both

**Table 5 | Difference in correlation coefficients between actual and model-fitted performance in the test phase for the Kalman:QU versus the Kalman:Q model in publicly available data sets 1 and 2**

| Comparison | r Kalman:QU | r Kalman:Q | Δ r | p value | 95% confidence interval of permutation distribution | Standardized effect size (SES) |
|---|---|---|---|---|---|---|
| Data set 1 | | | | | | |
| A1C1 | 0.785 | 0.704 | 0.081 | 0.120 | −0.088 0.087 | 1.338 |
| B1D1 | 0.721 | 0.498 | 0.223 | 0.002 | −0.141 0.148 | 2.962 |
| A2D2 | 0.812 | 0.702 | 0.109 | 0.006 | −0.078 0.078 | 2.762 |
| B2C2 | 0.768 | 0.286 | 0.482 | <0.001 | −0.201 0.197 | 4.742 |
| Data set 2 | | | | | | |
| A1C1 | 0.708 | 0.673 | 0.035 | 0.338 | −0.068 0.066 | 1.024 |
| B1D1 | 0.593 | 0.249 | 0.345 | <0.001 | −0.197 0.193 | 3.493 |
| A2D2 | 0.757 | 0.614 | 0.143 | 0.010 | −0.111 0.112 | 2.531 |
| B2C2 | 0.755 | 0.205 | 0.550 | <0.001 | −0.245 0.234 | 4.492 |

r is Pearson's correlation coefficient between actual and model-fitted learning performances. Two-tailed uncorrected p-values, confidence intervals, and standardized effects sizes were obtained from permutation tests ($n = 1000$ permutations).

partial- and complete feedback conditions[15,30,31], an interesting question is to what extent different decision strategies are applied as a function of different training regimes. On one hand, it would be efficient to apply only one strategy across all conditions, e.g., range-adaptation[15,30]. On the other hand, it may be more optimal to flexibly switch between context-based strategies, e.g., switching between strategies based on range-adaptation and estimation-uncertainty during respective complete and partial feedback conditions. This topic clearly needs to be addressed in future studies.

Obviously, estimation-uncertainty may not always be a critical component in the context of reinforcement learning, but this needs to be clearly demonstrated, for example by contrasting the fits of models with and without estimation-uncertainty components. Furthermore, the potential impact of estimation-uncertainty can be removed via careful experimental design, for example by providing feedback for both selected and rejected options, i.e., the complete feedback condition mentioned previously[15,30]. Another option, which also removes concerns regarding biased processing of feedbacks associated with rejected and selected options, is to use paradigms where the feedback is independent of the choice[32] and each option is presented equally many times[33]. These two manipulations respectively abolish the incentives for information-based exploration during learning and differences in estimation-uncertainty after the learning phase. Finally, estimation-uncertainty can be experimentally controlled by manipulating each option's sampling rate, such as in the 'Horizon' task where participants passively view stimulus-outcome associations before making their own decisions[6,7].

Beyond the main purpose of the study, it is interesting to note that Kalman-filter models outperformed the model with separate learning rates for better-then-predicted outcomes (positive prediction errors) and worse-than-predicted outcomes (negative prediction errors). This dual-learning rate model has been frequently used to demonstrate relationships between reinforcement learning biases and neurological and psychiatric conditions[21-23,34]. Given that a large amount of literature support dual-learning rate models, its inferior performance in the present study may seem surprising. However, because most previous studies did not include models based on Kalman-filtering, it is unclear to what extent such models would provide better fits to performance in those studies. In relation to this, it may be informative to consider why the Kalman-filter approach provides a better fit to performance. First, the Kalman-filter approach adjusts the rate of learning based on how much has been learned about a stimulus, i.e., the learning rate is initially rapid but slow during later stages of learning. By contrast, the learning rates of the dual-learning model are constant across the whole learning session. Accordingly, Kalman-filter models may more accurately represent the rate of learning as it dynamically changes across

the experiment. Second, because negative/positive prediction errors are relatively more prevalent during initial/later stages of learning, it could be argued that the dual-learning rate model is capable of capturing some variance associated with early vs. late learning. However, given their very definition, these learning rates also needs to account for differential learning associated with the valence of prediction error themselves (i.e., positive/negative prediction errors). In other words, learning rates in the dual-learning rate model may need to represent multiple aspects of learning behavior, which when uncorrelated decreases model performance.

Finally, our study provides predictions for future studies regarding the impact of increased exploration observed in loss (vs. gain) conditions[35-37]. Accordingly, because exploration during learning reduces the sampling bias between good and bad options, when tested after learning, conditions with high exploration rates should display a smaller bias between decision strategies related to expected value and estimation uncertainty. Specifically, good-good/bad-bad decisions with options drawn from gain conditions during learning (i.e., with less exploration) should respectively depend more on expected values/estimation uncertainty options. By contrast, this bias should be reduced for options drawn from loss conditions (i.e., with increased exploration), because good/bad options are sampled less/more frequently than in gain conditions.

## Methods

The study was performed in accordance with the Declaration of Helsinki and approved by the ethical review board at the Weizmann Institute of Science (IRB protocol: 1086-2).

### Participants

After having provided written consent according to the ethical regulations of the Weizmann Institute of Science, fifty-two participants took part in the experiment (27 self-reported males and 25 self-reported females; average age ± STD: 29.48 ± 8.36). All participants were native Hebrew speakers without any previous history of psychiatric or neurological disorders. Two participants were excluded because their average performance in four learning conditions were below chance-level.

### Reinforcement learning task

The experiment consisted of two phases, a learning phase and a subsequent test phase. All participants were unaware of the nature of the test phase (i.e., that they would be tested on the learned material).

**Learning phase.** During each trial in the learning phase, participants selected one of two options, and received feedback after each

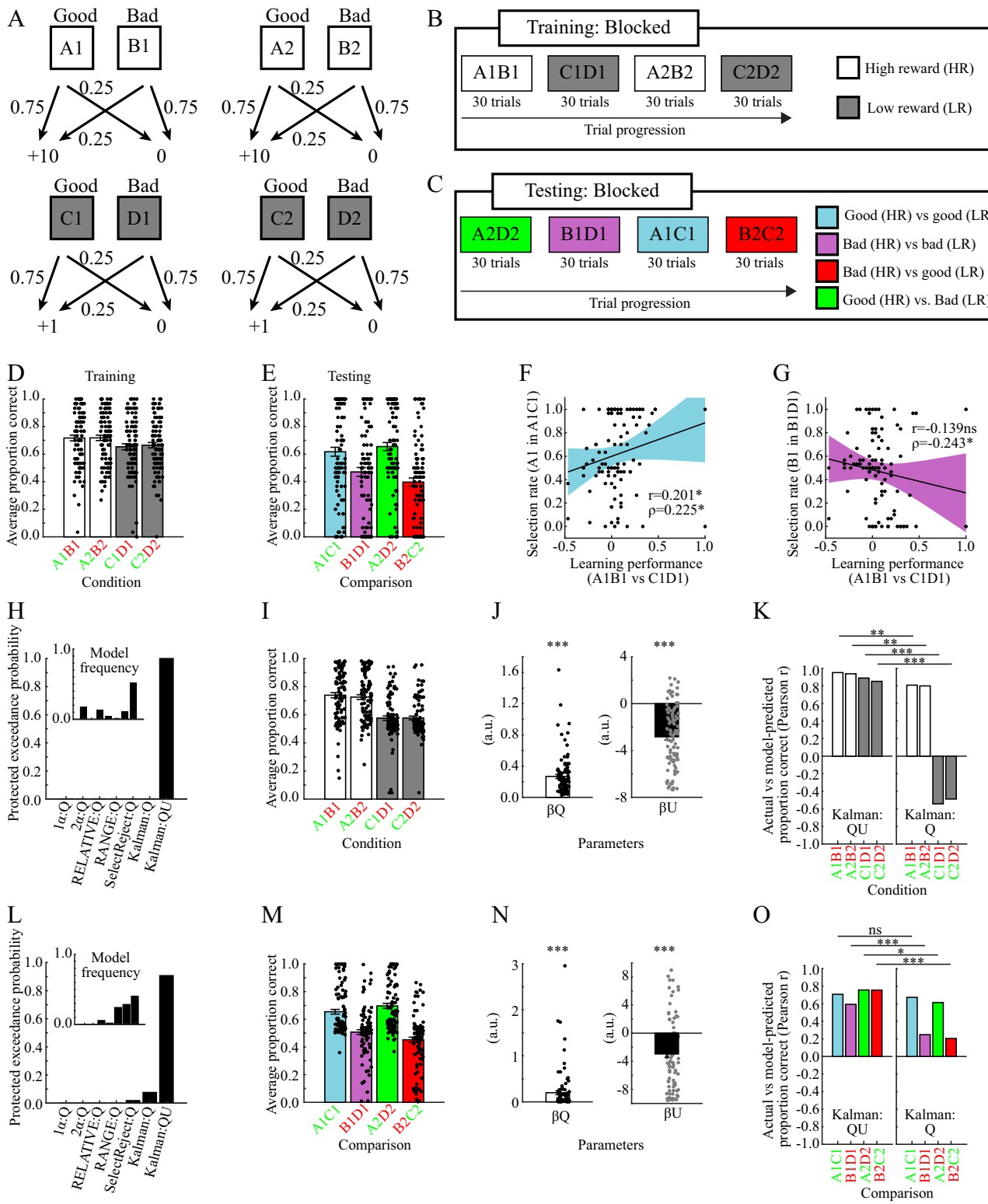

selection (Fig. 1A). A Good option in each pair provided positive/negative feedback with a probability of 0.75/0.25, while the other Bad option provided positive/negative feedback with a probability of 0.25/0.75.

The experiment contained five conditions, which differed in the probability of the feedback being drawn from an appetitive context (positive/negative feedback = +1₪/0₪), or an aversive context (positive/negative feedback=0₪/-1₪). For example, positive feedback in the most rewarded condition (i.e., $p_{Appetitive}$ = 1.0) was always +1₪, while it

was always 0₪ in the least rewarded condition (i.e., $p_{Appetitive}$ = 0.0). In three intermediate conditions, the probabilities that the feedbacks were drawn from an appetitive context were $p_{Appetitive}$ = 0.75, 0.5, and 0.25, respectively. Please observe that while the expected values are linearly dependent on $p_{Appetitive}$, the entropy is related to $p_{Appetitive}$ in an inverse-U-shaped fashion. Furthermore, the relative difference in expected values between Good and Bad options was identical across all conditions. The different conditions, including the expected values and the Shannon entropy for the Good and Bad options in each

**Fig. 5 | Validation 2 (n = 100). A** Schematic of stimulus-outcome contingencies **B** Schematic of trial progression during the learning phase. **C** Schematic of trial progression during the test phase. **D** Average actual learning performance. **E** Average actual test performance. **F** Positive correlation (Pearson, Spearman) between differences in learning performance and selection biases in the test phase for Good versus good pairs (r = 0.201, p = 0.045; ρ = 0.225, p = 0.025). **G** Negative correlation (Pearson, Spearman) between differences in learning performance and selection biases in the test phase for Bad versus Bad pairs (r = −0.139, p = 0.167; ρ = 0.243, p = 0.015). Please observe that the learning performance is inversely related to the sampling-rate of Bad options (i.e., sampling-rate of Bad options = 1-learning performance). **H** Protected exceedance probabilities and model frequencies (inset) for the learning phase. **I** Average model-fitted learning performance. **J** Fitted values for the model-parameters of the Kalman:QU model (t(49)$_{\beta Q}$ = 10.402, p < 0.001, 95%CI = [0.215 0.317], d = 1.040; t(49)$_{\beta U}$ = 10.404, p < 0.001, 95%CI = [−3.300 −2.243], d = 1.040). **K** Correlations (Pearson) between actual and model-fitted learning performance in each condition for the Kalman:QU (left panel) and the Kalman:Q model (right panel). The Kalman:QU model provides a better fit in all conditions. Δr$_{A1B1}$ = 0.143, p$_{Permutation}$ = 0.005, 95%CI$_{Permutation}$ [−0.104 0.098], SES = 2.823; Δr$_{A2B2}$ = 0.138, p$_{Permutation}$ = 0.005, 95%CI$_{Permutation}$ [−0.100 0.087], SES = 2.782; Δr$_{C1D1}$ = 1.433, p$_{Permutation}$ < 0.001, 95%CI$_{Permutation}$ [−0.356 0.371], SES = 7.953; Δr$_{C2D2}$ = 1.340, p$_{Permutation}$ < 0.001, 95%CI$_{Permutation}$ [−0.345 0.352], SES = 7.351). **L** Protected exceedance probabilities and model frequencies (inset) for the test phase. **M** Average model-fitted test performance. **N** Fitted values for the model-parameters of the Kalman:QU model (t(49)$_{\beta Q}$ = 4.551, p < 0.001, 95%CI = [0.110 0.280], d = 0.455; t(49)$_{\beta U}$ = 6.194, p < 0.001, 95%CI = [−3.907 −2.011], d = 0.619). **O** Correlations (Pearson) between actual and model-fitted performance in each comparison for the Kalman:QU (left panel) and the Kalman:Q model (right panel). The Kalman:QU model provides a better fit in all conditions including Bad options, but not the condition which includes two Good options (A1C1; blue bar; Δr$_{A1C1}$ = 0.035, p$_{Permutation}$ = 0.338, 95%CI$_{Permutation}$ [−0.068 0.066], SES = 1.024; Δr$_{B1D1}$ = 0.345, p$_{Permutation}$ < 0.001, 95%CI$_{Permutation}$ [−0.197 0.193], SES = 3.493; Δr$_{A2D2}$ = 0.143, p$_{Permutation}$ = 0.010, 95%CI$_{Permutation}$ [−0.111 0.112], SES = 2.531; Δr$_{B2C2}$ = 0.550, p$_{Permutation}$ < 0.001, 95%CI$_{Permutation}$ [−0.245 0.234], SES = 4.492). All errorbars indicate the standard error of the mean. d=Cohen's d. SES=Standardized effect size. p$_{Permutation}$ and 95%CI$_{Permutation}$ is the p-value and confidence interval for the null-distribution obtained via permutation testing. ***p < 0.001, **p < 0.01, *p < 0.05, ns not significant.

condition, are described in Supplementary Table 1. To clarify, the expected value of each option is the multiplication of p$_{Appetitive}$ with the respectively reward probabilities for each feedback type, e.g., for the Good option in the p$_{Appetitive}$ = 0.5 condition, the expected value is: 0.75*[0.5*(+1₪) + 0.5*(0₪)] + 0.25*[0.5*(0₪) + 0.5*(-1₪)] = 0.25₪. For the Bad option in the p$_{Appetitive}$ = 0.0 condition, the expected value is: 0.25*[0.0*(+1₪) + 1.0*(0₪)] + 0.75*[0.0*(0₪) + 1.0*(-1₪)] = −0.75₪.

The learning phase consisted of three blocks separated by self-paced breaks (Fig. 1B). Each block contained all of the five conditions, each presented for 16 consecutive trials (in total 80 trials per block). New objects were presented in each block, for a total of 15 different pairs of 30 objects. The order of conditions within each block was pseudo-randomized, such that the presentation order in blocks one and three were reversed, while it was randomized in the second block.

During a trial, the two objects were presented for 3.0 s., during which participants were required to make a response. The selected object was then highlighted and the objects remained on the screen for the remainder of the duration. Following a jittered delay, feedback was presented for 1.0 s., followed by an inter-trial-interval of a jittered delay. The jittered delay was drawn from a uniform distribution with range 0.5–7.5 s.

Task instructions for the learning phase were as follows:

"In each trial, select one of two objects to see how much it is worth. Some objects are more valuable than others. You may use the feedbacks (−1.00, 0.00, +1.00 NIS) to estimate the value of each object. The objects may not always provide the same outcome when selected. Your total score in NIS will be added to your compensation for this experiment."

To get familiarized with the task, all participants performed 16 trials each with the p$_{Appetitive}$ = 0.0 and p$_{Appetitive}$ = 1.0 conditions prior to starting the experiment proper. Accordingly, with regards to the game mechanics, the participants' knowledge was largely limited to knowing the possible feedback types (-1, 0, +1₪).

**Test phase.** In the post-learning phase participants selected one of two objects in each trial. To prevent further learning and incentives for exploration, no feedback was presented in this phase.

Three types of object pairs were presented. First, 'good versus good' pairs in which a Good object from one condition was paired with a Good object from another condition within the same block. For example, all Good objects from block one were paired yielding ten different pairs. A total of 30 new 'good versus good' pairs were created. Because each condition has a different p$_{Appetitive}$, the Good object from the higher p$_{Appetitive}$ condition is better because it has a higher expected value than the other Good option. Second, in a similar manner all Bad options were paired (and since the paired objects come from different conditions, one of them has a lower expected value and is therefore worse than the other option), giving a total of 30 new 'bad versus bad' pairs. Note that we only created pairs from objects within the same block to minimize the impact of different delays between the learning and the test phase (e.g., stimulus-outcome associations learned in the first block may be remembered differently as compared to associations learned in the third block)[17,18]. Finally, to ensure that participants were able to transfer learned information to the test phase, we also created 'good versus bad' pairs in which a Good object from one block was paired with Bad objects from the same condition but from a different block. In total, 30 such pairs were created.

For each trial, participants had 10.0 s to make a response, after which the selected object was highlighted for 0.5 s. The next trial was initiated after a jittered delay (uniform distribution with average delay of 1.0 s). In total, each new pair was presented two times for a total of 180 trials (30 'good versus good', 30 'bad versus bad', and 30 'good versus bad', each repeated twice). The trial order was randomized (Fig. 1C).

Task instructions for the testing phase were as follows:

"In each trial, try to select the most valuable of the two objects. Use what you learned in the previous phase of the experiment to guide your choices. No feedbacks will be shown during this phase of the experiment. Your total score in NIS will be added to your compensation for this experiment."

To get familiarized with the task, all participants performed eight trials with new combinations of objects presented during the training of the previous phase of the experiment.

Participants were paid for their time after the experiment, plus a monetary bonus consisting of the amount earned according to all their selections, hence providing motivation to both learn during the learning phase and to use the learned information during the test phase.

The main experiment was presented using Matlab together with Psychtoolbox (version 3.0.19)[38], and the images were obtained from the Snoddgrass and Vanderwart 'Like' Objects dataset[39], available under the Creative Commons Attribution-NonCommercial-ShareAlike 3.0 Unported License, and downloaded from the Tarr-lab homepage (https://sites.google.com/andrew.cmu.edu/tarrlab/).

**Statistical analysis.** Behavioral performances were assessed via a combination of standard statistical tests, such as two-tailed paired t-tests, Pearson and Spearman correlations, and repeated measures ANOVAs. In addition, linear mixed-effects models were used when adding condition-specific continuous covariates, such as sampling-

rates. Prior to each test, it was ensured that the data met the proper assumptions via the Kolmogorov-Smirnov test (normality)[40] and Bartlett's test (homoscedasticity)[41].

In a static environment, estimation-uncertainty is monotonically and negatively correlated with an options sampling-rate. Accordingly, the sampling-rate can be used to provide model-agnostic evidence for a relationship between estimation-uncertainty acquired during the learning and decision biases in the test-phase. Specifically, for a Good option the sampling-rate is exactly equal to the learning performance in the pair in which it was learned, while for the corresponding Bad option the sampling-rate is exactly equal to one minus that learning performance. Correlations between learning performances and decisions biases were calculated in a broad fashion (i.e., by collapsing an option's decision biases across all comparisons in the test-phase), and in a more fine-grained fashion (by testing pair-wise comparisons in the test-phase).

### Behavioral modeling

For clarity, we separate the descriptions of the learning part of the models from their decision parts. In brief, there are six different learning models and two different decision models. In the main text, each model is described by a learning part and a decision part, separated by a ':'. For example, a Kalman-filter learning model which makes decisions by considering both expected values and estimation-uncertainty is denoted as 'Kalman:QU'.

**Learning models.** Q-learning with one learning rate (1α model): First, we implement the canonical Q-learning model[42]. Here, the expected value $Q(t)_i$ of a selected option $i$ in trial $t$ is updated by a prediction error $\delta(t)_i$, which is the mismatch between the value of the received outcome $R$ and the selected object's expected value $Q(t)_i$, scaled by a constant learning rate $\alpha$:

$$Q(t+1)_i = Q(t)_i + \alpha * \delta(t)_i \tag{1}$$

$$\delta(t)_i = R - Q(t)_i \tag{2}$$

Here, $\alpha$ is a free parameter.

Q-learning with two learning rates (2α model): Behavioral biases expressed after learning may be due to differences in learning from positive and negative prediction errors[22,23]. To account for this possibility, a second model allowed different learning rates for positive and negative prediction errors:

$$Q(t+1)_i = Q(t)_i + \begin{cases} \alpha_+ * \delta(t)_i, \text{if } \delta(t)_i > 0 \\ \alpha_- * \delta(t)_i, \text{if } \delta(t)_i < 0 \end{cases} \tag{3}$$

The prediction error is defined as above (Eq. 2), while $\alpha_+$ and $\alpha_-$ are free parameters.

Q-learning of the selected and the rejected option (SelectReject model): Canonical Q-learning updates only the option that elicits a prediction error, i.e., the selected option in the present study. However, participants may use the feedback elicited by one option to update also the rejected option. Such a mechanism would allow participants to simultaneously learn the 'goodness' of the Good option and the 'badness' of the Bad option, something would could help explain observed decision biases in the current study[43]. Accordingly, a model was designed which updates the rejected option's expected value with the reverse-signed prediction error of the selected option:

$$Q(t+1)_i = Q(t)_i + \begin{cases} \alpha_{Selected} * \delta(t)_{Selected}, \text{if } i = Selected \\ \alpha_{Rejected} * -\delta(t)_{Selected}, \text{if } i = Rejected \end{cases} \tag{4}$$

$$\delta(t)_{Selected} = R - Q(t)_{Selected} \tag{5}$$

Here, $\alpha_{Selected}$ and $\alpha_{Rejected}$ are free parameters.

Two other learning models were implemented because they have previously been used to explain behavior in similar tasks, i.e., tasks with a learning and a test phase, but with a focus on the contextual scaling of feedback values[15,30]. In general, both of these models suggest that feedback values are scaled based on expected outcome values of the context in which the feedback is received. These models provide explanations as to why Good options learned in a highly rewarding learning context are not always selected more frequently than Good options learned in less rewarding contexts, when tested in a post-learning test phase[15,30].

Contextual centering of outcomes (RELATIVE model): The RELATIVE model (also referred to as the REFERENCE model[31]) adjusts the outcome value based on the context value in that trial, $V(t)_c$, where each context $c$ is a pair of objects[30]:

$$Q(t+1)_i = Q(t)_i + \alpha * \delta(t)_i \tag{6}$$

$$\delta(t)_i = R - V(t)_c - Q(t)_i \tag{7}$$

The context value V in context $c$ is updated as:

$$V(t+1)_c = V(t)_c + \alpha_c * \delta(t)_c \tag{8}$$

$$\delta(t)_c = \frac{(R + Q(t)_{Rejected})}{2} - V(t)_c \tag{9}$$

Here, R is the outcome of the selected option, while $Q(t)_{Rejected}$ is the expected value for the rejected option. The learning rate $\alpha_c$ scales the prediction error of the context $\delta(t)_c$, while $\alpha$ scales the prediction error for the selected option $i$, i.e., $\delta(t)_i$. In essence, this model adjusts outcome values based on an estimate of the average outcome value within a context. Accordingly, a positive outcome value (e.g., +5) can actually be perceived as negative in a highly rewarding context (e.g., whenever $V_c$ > +5).

Contextual normalization of outcomes (RANGE model): The RANGE model normalizes reward outcomes based on the range of available feedbacks in a context $c$[15]:

$$Q(t+1)_i = Q(t)_i + \alpha * \delta(t)_i \tag{10}$$

$$\delta(t)_i = R_{RAN} - Q(t)_i \tag{11}$$

Where the relative reward $R_{RAN}$ is calculated as:

$$R_{RAN}(t) = \frac{R_{OBJ}(t) - R_{MIN}(t)_c}{1 + R_{MAX}(t)_c - R_{MIN}(t)_c} \tag{12}$$

Where $R_{OBJ}$ refers to the received feedback and the maximum ($R_{MAX}$) and minimum ($R_{MIN}$) reward values of a context being updated as:

$$R_{MAX}(t+1)_c = R_{MAX}(t)_c + \alpha_R * (R_{OBJ}(t) - R_{MAX}(t)_c), \text{if } R_{OBJ}(t) > R_{MAX}(t)_c \tag{13}$$

$$R_{MIN}(t+1)_c = R_{MIN}(t)_c + \alpha_R * (R_{OBJ}(t) - R_{MIN}(t)_c), \text{if } R_{OBJ}(t) < R_{MIN}(t)_c \tag{14}$$

Therefore, in essence the RANGE model gradually learns the maximal ($R_{MAX}$) and the minimal ($R_{MIN}$) outcome values in a context, and normalizes the actual feedback ($R_{OBJ}$) based on these limits. Practically, this ensures that expected values are scaled between 0 and 1, thus providing a more general approach to contextual scaling than the RELATIVE model.

Kalman-filter learning model (Kalman model): The Kalman-filter approach uses dynamic learning rates and provides an intuitive way to

estimate estimation-uncertainty, which is monotonically and inversely correlated with an options sampling-rate. The Kalman-filter approach has been used to describe learning behavior in similar n-armed bandit tasks with normally distributed outcomes[8,9]. Importantly, while the Kalman-filter is an optimal linear estimator for normally distributed process errors and noise[44], it can readily be applied to other types of distributions[45,46].

Value updating is similar to Q-learning, with the exception of a dynamic learning rate which is specific for each object and depends on its current level of estimation-uncertainty:

$$Q(t+1)_i = Q(t)_i + \alpha(t)_i * \delta(t)_i \qquad (15)$$

The prediction error $\delta(t)_i$ is defined as above (Eq. 2).

The estimation-uncertainty of object $i$ is estimated by the standard deviation $\sigma(t)_i$, which is updated according to:

$$\sigma(t+1)_i^2 = \sigma(t)_i^2 - \alpha(t)_i * \sigma(t)_i^2 \qquad (16)$$

The learning rate $\alpha(t)_i$ for object $i$ in trial $t$ is given by:

$$\alpha(t)_i = \frac{\sigma(t)_i^2}{\sigma(t)_i^2 + \sigma_0^2} \qquad (17)$$

Q-values were initialized as the expected value of a randomly selected option given three possible outcomes $r$, their respective magnitudes $R(r)$ and probabilities $p(r)$: $\sum_r R(r)*p(r)$, where $p(r) = 1/3$. The constant $\sigma_0$ was calculated as the standard deviation of the categorical distribution given $R(r)$ and $p(r) = 1/3$:

$$\sigma_0 = \sqrt{\sum_r R(r)^2 * p(r) - \left(\sum_r R(r) * p(r)\right)^2} \qquad (18)$$

For the Kalman-filter learning models, the initial estimation-uncertainty was set to $\sigma_0$.

All learning rates (i.e., all $\alpha$) were constrained between 0 and 1, while all decision weights (i.e., all $\beta$) were constrained between −20 and 20. To ensure that the results are not caused by using too strict priors, we allowed the fitted parameters to vary widely across both individual and group-level parameter-fits (i.e., for individual fits, the prior means for all $\alpha$ and all $\beta$ were set to 0.5 and 0, respectively, while the variance for all fits was set to 16.25).

**Decision models.** Decisions were modeled using the softmax choice probability function, where the probability of selecting option $i$ in trial $t$ (rather than the other option j) is estimated as:

$$p(t)_i = e^{u(t)_i} / (e^{u(t)_i} + e^{u(t)_j}) \qquad (19)$$

$u(t)_i$ is the utility of option $i$ in trial $t$, which is calculated as the sum of each behavioral factor's value scaled by their respective decision weights. Here, two factors were considered to influence the utility, namely the expected values (i.e., the Q-values), which was provided by all learning models, and the estimation-uncertainties (i.e., the standard deviation $\sigma$), which was provided by the Kalman-filter learning model.

Decision model Q: The utility for decision model Q:

$$u(t)_i = Q(t)_i * \beta_Q \qquad (20)$$

Decision model QU: The utility for decision model QU:

$$u(t)_i = Q(t)_i * \beta_Q + \sigma(t)_i * \beta_U \qquad (21)$$

The $\beta$-values are free parameters which determine the impact of a particular factor to a decision. For example, a large negative $\beta_U$ indicates that estimation-uncertainty is perceived as highly aversive, while a large positive $\beta_Q$ indicates that decisions are strongly biased in favor of expected-values.

## Model fitting

All models were separately fit to behavior during the learning phase and the test phase, such that two sets of model-parameters were obtained for each phase of the experiment. To fit the models to behavior, we used a hierarchical Bayesian inference (HBI) method[16]. The HBI concurrently fits parameters and compares the considered models, something which allows constraining individual fits to group-level hierarchical priors. Additionally, the random effects approach used by the HBI calculates both group-level statistics and model evidence based on the posterior probability that the model explains each participants's choice data. The HBI method provides fitted model parameters for each participant, as well as protected exceedance probabilities for each model. The exceedance probability estimates the probability that a model is the most likely model to explain the observed behaviors, as compared to all other considered models[47]. The *protected* exceedance probability is more conservative, by also taking into account the possibility that none of the compared models is supported by the data[48]. Importantly, the HBI method provides a superior identification of models as compared to other commonly used hierarchical and non-hierarchical procedures[16]. Further, the exceedance probability provides a robust measure of model-selection which is not sensitive to factors, such as outliers, which are known to bias the model-selection when using fixed-effect procedures (e.g., the group Bayes factor) or frequentist tests (e.g., t-tests)[47].

## Model simulation

**Model recovery.** To ensure that different models give rise to distinguishable behaviors[49], for each model we generated behaviors of 200 virtual participants by randomly selecting model-parameter values across a broad range (e.g., uniformly distributed within the range of possible values). Next, all models were fitted to these virtual-behavior datasets using the same HBI procedure described previously, and the resulting model frequencies were used to calculate a confusion matrix indicating how likely each model was to having generated the simulated behavior of each model (see below). This procedure is straightforward for the learning phase, but requires an additional step when simulating test-phase behaviors. For this reason, we first generated the behavior of 200 virtual participants in the learning phase using the most parsimonious model for the learning phase. These behaviors were then used as input when simulating test phase behaviors. In other words, for each model the behavior of 200 virtual participants were generated using a broad range of model-parameter values, all models were fitted to these behaviors, and finally the model frequencies and the confusion matrix were calculated.

To clarify, the output of the model-recovery procedure is a confusion matrix, where the proportion of model-selections for each fitted model (y-axis) and each generating model (x-axis) is reported. For perfect model-recovery all diagonal entries are 1, indicating that behaviors generated by a generating model is best fitted by that generating model. In other words, each model gives rise to unique behavioral patterns. However, at the very least, the generating model should be the most frequently selected model[50].

**Parameter recovery.** To ensure that the parameters included in a model are meaningful, e.g., do not share variance with other parameters, the parameter values used to generate simulated behaviors need to be successfully recovered when fitting the generating model to the simulated behaviors[49]. Accordingly, we randomly selected values of model parameters for the most parsimonious model (within the range of the fitted values), and generated the behavior of 200 virtual participants. Next, the same model was re-fitted to the simulated behaviors, and correlation coefficients were calculated between the

generating and the recovered parameters. This analysis was performed for both the learning and the test phase of the first experiment. Ideally, correlations between the same parameters should be strongly positive, with weak or no correlations between different parameters.

**Model validations.** To ensure that the best model not just provides the best fit relative other models, but also resembles actual behavior, we plot model-fitted behaviors, calculate inter-individual correlations between actual and model-fitted behavior, and provide McFadden's pseudo $R^2$ for the winning models:

$$R^2 = 1 - \frac{LL_{Model}}{LL_{Null}} \qquad (22)$$

$LL_{model}$ is the log-likelihood of the model and $LL_{Null}$ is the log-likelihood of a 'null' model without free parameters.

Furthermore, we compared model-fitted behaviors between models using permutation tests, where first, the actual difference in correlation coefficients between model-fitted and actual behavior was calculated ($\Delta r$actual). Then, a null-distribution ($\Delta r$permuted) was created by shuffling the actual behaviors, calculating a new $\Delta r$, and repeating this step for 1000 times. A two-tailed p-value could be obtained by comparing $\Delta r$actual with the null-distribution: $p = mean(|\Delta r_{permuted}|) > |\Delta r_{actual}|$. 95% confidence intervals were obtained by calculating the 2.5% and 97.5% percentiles of the null distribution, and the standardized effect size (SES) was calculated as:

$$SES = \frac{(\Delta r_{actual} - mean(\Delta r_{permuted}))}{std(\Delta r_{permuted})} \qquad (23)$$

**Public datasets**

To confirm our findings, and to test their generalizability across behavioral paradigms, we re-analyzed data obtained in a publicly available datasets generously provided online (published by Bavard, Rustichini, and Palminteri)[15]. Importantly, while this dataset contains eight different experiments, we re-analyzed only those matching our experiment. Specifically, necessary inclusion criteria were: (i) the training should be conducted with partial feedback (to enable biases in sampling rates and estimation-uncertainty), and (ii) the test phase should be conducted without feedback (to confirm the main hypothesis regarding the impact of estimation uncertainty on decision making in the absence of learning). Two experiments fulfilled these criteria (described below), while four experiments were excluded because complete feedback was provided during training, and two other experiments were excluded because feedback was provided during the test phase. Because their methods and results have been described extensively elsewhere[15], here we provide only a brief overview of their experiments using our own denotations.

In these experiments, participants first learned to select the Good option in four different pairs (denoted by A's and C's in Figs. 3A, 4A) over their corresponding Bad option (denoted by B's and D's). The reward probabilities for the Good and Bad options were respectively 0.75 and 0.25 (Figs. 3A, 4A). In two pairs (A1B1 and A2B2), the reward was large (+10 points) while in the two other pairs it was small (+1 point; C1D1 and C2D2). Non-rewarding feedback was always 0. For the post-learning decision phase, four new pairs were created: a 'good versus good' pair (A1 versus C1), a 'bad versus bad' pair (B1 versus D1), and two mixed pairs (A2 versus D2; B2 versus C2). The pairs were presented in an inter-mixed fashion in their 'Experiment design E1' (Fig. 3B, C), while in their 'Experiment design E5' the pairs were presented block-wise during both learning and testing (Fig. 3B, C). During the learning and the test phase, each pair was presented 30 times.

**Reporting summary**

Further information on research design is available in the Nature Portfolio Reporting Summary linked to this article.

## Data availability

The data from the main experiment is publicly available in an Open Science Framework repository[51], https://doi.org/10.17605/OSF.IO/3DK95. Data from the two complementary analyses are available in a Github repository[15], https://github.com/hrl-team/range. Source data are provided with this paper.

## Code availability

Custom code used to fit behavioral models and to simulate behavior is publicly available in an Open Science Framework repository[51], https://doi.org/10.17605/OSF.IO/3DK95.

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

## Acknowledgements

K.C.A. is the incumbent of the Sam and Frances Belzberg Research Fellow Chair in Memory and Learning. The work was supported by an ISF #1468/24 grant to R.P. and by the Drescher Center for Research on Mental and Emotional Health. We are sincerely grateful to Sasha Devore for insights contributed at various stages of the project. Stimulus images courtesy of Michael J. Tarr, Carnegie Mellon University, http://www.tarrlab.org/.

## Author contributions

K.C.A. designed the experiment, analyzed the data, and wrote the paper. L.A. collected and analyzed the data. R.P. designed the experiment, analyzed the data, and wrote the paper.

## Competing interests

The authors declare no competing interests.
