## [Transparent Peer Review file · Nature Communications]

Estimation-uncertainty affects decisions with and without learning opportunities

Corresponding Author: Dr Kristoffer Aberg

Version 0:

Reviewer comments:

Reviewer #1

(Remarks to the Author)

Review of "Better the Devil You Know" by Åberg et al.

The authors investigate the role of "estimation uncertainty" in preference formation during reinforcement learning in an experiment organized with a learning/transfer structure involving a multi-bandit task. They support their findings with a combination of clever behavioral and computational analyses. The study is well-designed, and they conduct several important sanity checks on model-based results. Additionally, they re-analyze a subset of experiments from Bavard et al. (2021; B2021), which appear to confirm their findings. I appreciate the research question and the overall study; however, I have some points of concern.

Presentation Issues

1/ I found the abstract and introduction somewhat unclear. The main issue is that they rely too heavily on the assumption that readers will immediately understand terms like "estimation error" and, even more confusingly, "information gain" in the context of this study. Even as someone familiar with multi-bandit tasks, I was unsure of the study's direction after my first read of the abstract and introduction. Crucially, the concept of sampling bias, which is key to understanding the study, is introduced later in the text. I suggest restructuring these sections to begin by explaining how reinforcement learning (RL) can suffer from sampling bias and why this is problematic, as it can lead to estimation error.

The authors mention that sampling bias is "outside the control of the experimenter." While I understand this perspective, it isn't entirely accurate. There are experimental manipulations that can eliminate (complete feedback) or reduce (by ordering outcomes specifically) this bias.

2/ I found most of the figures difficult to interpret, especially the unconventional presentation of conditions in Figure 1A. Additionally, the fact that the second figure contains so many panels (A-R) is problematic. More importantly, it was challenging to infer transfer phase preferences from the graphs, as well as to understand how simulations compared to actual data. Palminteri's and Hayes's groups have developed several clear and accessible ways to display transfer test results, which the authors could leverage for better visualization.

To recap, ideally, the figures should convey participants' preferences and clearly show how individual models perform in simulations.

Substantive Issues

1/ I am only partially convinced by the authors' narrative and the generalizability of their computational results. A more serious challenge to their claims lies in their re-analysis of only two of the eight experiments from B2021. While they do not clearly justify their selection, I assume they chose experiments with 1) partial feedback (since complete feedback eliminates sampling bias) and 2) no feedback during the transfer phase (presumably to match their own design). They claim that their model provides a better account of the data. However, I am unsure how a model lacking a range component can explain the absence of a magnitude effect in the learning phase of B2021.

2/ Furthermore, in B2021, context-dependence is demonstrated by preference for C in BC choices during the transfer test. The authors argue (reasonably) that this could be due to limited sampling of option B during the learning phase. If this is the

case, the effect (preference for C in BC) should disappear in complete feedback experiments. Yet, the opposite occurs: in B2021, as well as in Bavard et al. (2018) and Palminteri et al. (2015), the preference for C in BC choices is stronger in complete feedback tasks—those where there is no sampling bias, and “estimation uncertainty” should play no role. Thus, their process may explain this context-dependent effect only in partial feedback experiments, but not in complete feedback ones. This leads to a somewhat convoluted explanation, requiring different computational accounts to explain the same effect under different conditions.

3/ I am also uncertain about the authors' rejection of dual learning-rate models. These models are a robust feature of human reinforcement learning, even in complete feedback situations. While the authors rely on their model comparison results, what about the substantial body of work supporting dual learning rates? This is not merely an academic question, as distorted values from alpha+/alpha- models might also contribute to an estimation uncertainty effect. For instance, if one overestimates the value of the chosen option optimistically, it could appear as though they are underestimating the unchosen one.

Minor Points

1/ the title is also debatable

2/ I suggest renaming the RELATIVE model as the REFERENCE model, as the concept of outcome relativity also applies to the RANGE model.

see doi.org/10.1016/j.cobeha.2021.06.006

Reviewer #2

(Remarks to the Author)

This manuscript describes one original experiment and two reanalyses of publicly available datasets to evaluate the hypothesis that how often an option is sampled and an observation observed in the context of an instrumental choice task (i.e. estimation uncertainty) influences choice preferences beyond learning and valuation. The purpose of the original experiment was to setup learning contexts in sampling of choice options was deliberately yoked to learning different rates of reinforcement. In doing so the authors could evaluate how learning related performance and estimation uncertainty influenced choice preferences after learning and across multiple reinforcement schedules during an unexpected choice test in which participants had to choose between options in terms of learned valuations. The main result relies on use of model comparisons to infer that learned avoidance of options associated with punishment is at least in part due to sampling those options less. This is shown in both the original experiment, and demonstrated again by applying the modeling approach to the two published datasets that used a similar learn and test task design. Overall I think this an important point that is being made by the authors in a straightforward manner. I have a few concerns:

1. One criticism is that they do not directly evaluate their claims by testing them in a task where sampling of good and bad options is equalized (e.g. through forced choice trials. Instead their claims rely entirely on demonstrating increased predictability of choices in the test scenario when comparing Kalman filter models that include or exclude estimation uncertainty. More discussion of the potential limitations of the modeling approach is warranted.
2. There is no discussion of the recent and relevant work by Anne Collins and colleagues examining the dissociable contributions of reinforcement learning and working memory to choice. There is a direct intersection between sampling and working memory, in that items sampled less often will be more prone to being forgotten and choice relationships more difficult to recall. To that end, and because the post-learning choice tests were unannounced, do the authors see differential effects of estimation uncertainty when choosing between pairs of bad or good options based on the block order in which learning occurred? Specifically, we're more recently learned options more or less susceptible to the influence of estimation uncertainty?
3. Considering that learning and estimation uncertainty were yoked in the original experimental design and the evidence that learning was weaker in the probabilistic conditions, do the authors claims hold if the deterministic learning conditions are excluded from the analyses? In these learning situations the subjects would only need to receive deterministic feedback in a limited number of trials before they could infer the value of both options presented (one good and one bad). I'm curious as to whether any self-reports were collected from the subjects indicating they were aware or not of the yoked reinforcement contingencies, or if there was behavioral evidence to indicate condition dependent differences in when they stopped sampling bad options. For example, I doubt the participants were showing matching behavior but in such a scenario estimation uncertainty should have much less influence, correct?

Reviewer #3

(Remarks to the Author)

This manuscript reports on the effects of expected value and uncertainty (specifically, estimation uncertainty) on choice behavior in a decision-making task. By using a specific task design – where different pairs of stimuli, learned in separate blocks, have the same value difference but different absolute expected values, and then assessing choices between stimuli from different blocks in a separate test phase – the authors are able to separately assess effects of expected value and unexpected uncertainty on choice, even after learning. Overall, this is a well-done and interesting paper. Most of my comments are targeted towards clarifying and strengthening the findings.

1. The discussion is rather superficial and mostly restates the results, and the end of the introduction is also weak. In particular, the manuscript does not discuss implications of the findings for theories of decision-making. Although I'm sure the word count is partially responsible, bringing in the following areas, if possible, would strengthen the manuscript:
 - a. Effects of losses on exploration: studies have found asymmetries in exploratory behavior when avoiding losses vs. seeking gains (e.g., Krueger, Wilson, & Cohen, 2017; Lejarraga & Hertwig, 2017). Do the findings in this paper help explain these effects?
 - b. Implications of the Kalman-QU model explaining the data better than other models
2. Related to 1b above, does the addition of the U part of the choice model to the non-Kalman models improve prediction?
3. The methods are well done – good model checking and use of model-free and model-based method. I have a few optional suggestions:
 - a. I was surprised to not see (or perhaps I missed) more interpretation of the parameters from the computational model – in particular, that participants are indeed uncertainty averse during both learning and test phases, but especially so (possibly significantly so?) in test relative to learning phases. The reporting of the ANOVA results was lengthy and it was more difficult to see the relation between those results and the overall goal of the paper.
 - b. In addition, in the results section, when discussing post-learning (test phase) decisions, it would be helpful to briefly review what learning performance is and its relation to outcome uncertainty. Otherwise the analyses can be confusing to follow.
4. Minor: the methods state each block has 90 trials, but should this be 80 trials (16 trials per condition x five conditions)?
5. For figure 2D, it may be helpful to sort the x-axis comparisons by entropy or otherwise highlight that most comparisons with the largest difference between good and bad are those with two high-entropy choices.

Krueger, P. M., Wilson, R. C., & Cohen, J. D. (2017). Strategies for exploration in the domain of losses. *Judgment and Decision Making*, 12(2), 104-117.

Lejarraga, T., & Hertwig, R. (2017). How the threat of losses makes people explore more than the promise of gains. *Psychonomic bulletin & review*, 24, 708-720.

Version 1:

Reviewer comments:

Reviewer #1

(Remarks to the Author)

Review for the Revision of Aberg et al. ("Better the Devil..." etc.)

I must express some slight disappointment with the generally low level of engagement and the limited modifications made by the authors in response to my suggestions. Some of my comments were dismissed as a "matter of taste." But I find it difficult to justify presenting a 16-panel figure as an effective and clear way to communicate key results. Similarly, the utility of an exceedance probability figure is questionable (as it does not show actual data points). That being said, I will focus my comments on the authors' responses to my substantive points 1 and 2.

1/ Selection of Experiments in Bavard et al. 2021

While the authors justify their selection of experiments to re-analyze based on their similarity to their own, the selection remains confounded by the omission of experiments where their model would fail. This may raise in other readers concerns about cherry-picking supporting examples from the literature. I recommend that the authors be extra-transparent about this and include analyses of the other experiments from Bavard et al. 2021. This would facilitate a more honest discussion of the points at issue. After all, the authors themselves concede that different cognitive processes explain different outcomes in different tasks, so why not show the full picture?

2/ Magnitude Effect

I could have been more precise in my initial review. While it is true that a significant magnitude effect is reported in Bavard et al. 2021, its small size is inconsistent with absolute value encoding and instead requires a range normalization process. Additionally, in a recent study (Anllo et al. 2024) using a similar task, the magnitude effect disappears entirely. My question to the authors is: Does their uncertainty avoidance model account for the very small magnitude effect observed during the learning phase of Bavard et al. 2021? Without resorting to ad hoc modifications (e.g., different temperatures for different pairs of options), I suspect it cannot. If this is the case, the authors may find themselves in the uncomfortable position of needing two different explanations for the learning effects (small magnitude effect) and transfer phases result (preference for C in BC).

3/ Task-Specific Explanations

The authors argue that their model explains key behavior in some tasks (partial feedback) but not others (complete feedback). While this position is tenable, it risks disappointing readers who expect models to offer general and universal explanations. By rejecting range normalization—which works across all tasks—in favor of task-specific explanations, the authors are, in a broader sense, giving up parsimony.

Moreover, this interpretational challenge becomes more significant in light of evidence showing that the key effect (preference for C in BC comparisons) is stronger in complete feedback experiments, even within-subject (see Palminteri et al. 2015; Nahuel-Salem 2023). These studies show that the same individuals exhibit stronger inversions in complete feedback conditions compared to partial feedback. Are the authors not surprised we require different explanations for two

very similar behavioral effects occurring within the same subject at the same time?

Reviewer #2

(Remarks to the Author)

The authors have adequately addressed my prior concerns.

Reviewer #3

(Remarks to the Author)

I thank the authors for their responses. I have no further comments

Version 2:

Reviewer comments:

Reviewer #1

(Remarks to the Author)

I have carefully read the authors' response, and I thank them for engaging with my comments in a thorough manner. While I still believe there are several points of disagreement regarding what constitutes a successful model and the appropriate burden of evidence for proposing and supporting new behavioural models, their paper should nonetheless provide an interesting contribution to these ongoing debates.

Reviewer #2

(Remarks to the Author)

My concerns were all addressed in the prior revision.

We sincerely thank the reviewers for their careful and insightful reviews of our manuscript, which have significantly increased its quality. Below, we address each of their concerns in a point-by-point fashion. Reviewers' comments are displayed in black font, while our responses are in blue font.

REVIEWER COMMENTS

Reviewer #1 (Remarks to the Author):

Review of “Better the Devil You Know” by Åberg et al.

The authors investigate the role of “estimation uncertainty” in preference formation during reinforcement learning in an experiment organized with a learning/transfer structure involving a multi-bandit task. They support their findings with a combination of clever behavioral and computational analyses. The study is well-designed, and they conduct several important sanity checks on model-based results. Additionally, they re-analyze a subset of experiments from Bavard et al. (2021; B2021), which appear to confirm their findings. I appreciate the research question and the overall study; however, I have some points of concern.

Presentation Issues

1/ I found the abstract and introduction somewhat unclear. The main issue is that they rely too heavily on the assumption that readers will immediately understand terms like “estimation error” and, even more confusingly, “information gain” in the context of this study. Even as someone familiar with multi-bandit tasks, I was unsure of the study’s direction after my first read of the abstract and introduction. Crucially, the concept of sampling bias, which is key to understanding the study, is introduced later in the text. I suggest restructuring these sections to begin by explaining how reinforcement learning (RL) can suffer from sampling bias and why this is problematic, as it can lead to estimation error.

Thank you for these suggestions. We have reworked the Abstract and the Introduction based on your suggestions. Most specific changes:

- The issue of sampling bias is introduced earlier in both the Abstract (line 13-15) and in the Introduction (line 35-37).
- The concept of information gain was removed from the Abstract, but is now clearly defined in the Introduction (line 49-52).

The authors mention that sampling bias is “outside the control of the experimenter.” While I understand this perspective, it isn’t entirely accurate. There are experimental manipulations that can eliminate (complete feedback) or reduce (by ordering outcomes specifically) this bias.

Thank you for this comment. We discuss this topic in the Discussion (lines 859-869), but removed it from the Introduction because we deemed it to be unnecessary there.

2/ I found most of the figures difficult to interpret, especially the unconventional presentation of conditions in Figure 1A. Additionally, the fact that the second figure contains so many panels (A-R) is problematic. More importantly, it was challenging to infer transfer phase preferences from the graphs, as well as to understand how simulations compared to actual data. Palminteri's and Hayes's groups have developed several clear and accessible ways to display transfer test results, which the authors could leverage for better visualization.

Thank you for these comments, which we address in a point-wise manner below:

Figure 1A: We have to disagree with the reviewer. We sincerely believe our presentation is the most straight-forward way to represent the different conditions of the task. It follows the conventional presentation of a probabilistic reinforcement learning task, with one added layer to present the addition of the different feedback-pools. Furthermore, because Table 1 was added to provide extensive information regarding the different conditions, we opted to keep Figure 1A as it is. If the reviewer strongly disagrees, we would be grateful for guidance on how to make Figure 1A clearer.

A large number of panels: This is perhaps more of a personal preference, but we believe all panels provide relevant information which should not be moved to a Supplementary section. Furthermore, for consistency and readability, we wanted to keep all results of an experimental phase in one Figure, which is why Figure 2 was not split into separate Figures showing the behavioral and modeling results. Accordingly, we decided to keep Figure 2 as is, but would be willing to split it if the reviewer insists.

Transfer results and simulations: If we understand correctly, the reviewer is specifically referring to Figures 2D, H, N-R? We initially created heatmaps to display selection biases, which is what the groups of Palminteri and Hayes frequently use to display similar results (i.e. selection biases between multiple different combinations of stimuli). However, after careful considerations we decided that the line-plots provide the best way to display our results. If the reviewer insists however, we will replace them with the heatmaps.

To recap, ideally, the figures should convey participants' preferences and clearly show how individual models perform in simulations.

We sincerely believe that our Figures clearly show participants' preferences, as well as the relationship between actual performance and model simulations. In particular, for the simulations we show the results of the parameter- and model recovery procedures, group-level performances, as well as inter-individual correlations between model- and actual performance for the relevant models.

Substantive Issues

1/ I am only partially convinced by the authors' narrative and the generalizability of their computational results. A more serious challenge to their claims lies in their re-analysis of only two of the eight experiments from B2021. While they do not clearly justify their selection, I assume they chose experiments with 1) partial feedback (since complete feedback eliminates sampling bias) and 2) no feedback

during the transfer phase (presumably to match their own design). They claim that their model provides a better account of the data. However, I am unsure how a model lacking a range component can explain the absence of a magnitude effect in the learning phase of B2021.

Thank you for demanding this highly relevant justification. Indeed, we selected the experiments based on two critical criteria related to our own design:

- Partial feedback during training (to allow the induction of sampling biases)
- No feedback during the test phase (to allow testing for lingering effects of estimation-uncertainty without further learning or need for exploration)

We now provide explicit justifications for including the two experiments in our re-analysis, as well as clear motivations why the other experiments were excluded (lines 398-406).

For the latter question, regarding the absence of a magnitude effect in the learning phase of B2021, we are not exactly sure what the reviewer refers to. There were significant magnitude effects reported during the learning in B2021:

“Replicating previous findings, in the learning phase, we also observed a moderate but significant effect of the choice contexts, where the correct choice rate was higher in the $\Delta EV = 5.0$ compared to the $\Delta EV = 0.5$ contexts (0.71 ± 0.18 versus 0.67 ± 0.18 ; $t(799) = 6.81$, $P < 0.0001$, and $d = 0.24$; Fig. 2C (6).”

Furthermore, while the authors did not report specific learning differences between the pairs within each experiment, we double-checked the two experiments analyzed in our manuscript and found all differences between each high EV pair (A1B1 or A2B2) versus each low EV pair (C1D1 or C2D2) within each experiment to be significant (all p-values < 0.05).

If we misunderstood the reviewers concern, we apologize and would be happy for a clarification.

2/ Furthermore, in B2021, context-dependence is demonstrated by preference for C in BC choices during the transfer test. The authors argue (reasonably) that this could be due to limited sampling of option B during the learning phase. If this is the case, the effect (preference for C in BC) should disappear in complete feedback experiments. Yet, the opposite occurs: in B2021, as well as in Bavard et al. (2018) and Palminteri et al. (2015), the preference for C in BC choices is stronger in complete feedback tasks—those where there is no sampling bias, and “estimation uncertainty” should play no role. Thus, their process may explain this context-dependent effect only in partial feedback experiments, but not in complete feedback ones. This leads to a somewhat convoluted explanation, requiring different computational accounts to explain the same effect under different conditions.

Thank you for this insight. We certainly agree that our model cannot explain behavioral differences in complete feedback conditions, at least given the presumption that the feedbacks for the selected and the rejected options are attended and/or processed in an unbiased fashion. However, we do not see a problem that different decision strategies are used. On the contrary, it seems quite reasonable that individuals switch between strategies based on current task demands.

To clarify the specificity of our findings to partial-feedback conditions, and to raise the issue of applying one model to all conditions versus switching between different ones, we added a paragraph to the Discussion (lines 844-856).

3/ I am also uncertain about the authors' rejection of dual learning-rate models. These models are a robust feature of human reinforcement learning, even in complete feedback situations. While the authors rely on their model comparison results, what about the substantial body of work supporting dual learning rates? This is not merely an academic question, as distorted values from alpha+/alpha- models might also contribute to an estimation uncertainty effect. For instance, if one overestimates the value of the chosen option optimistically, it could appear as though they are underestimating the unchosen one.

Thank you for raising this issue. We are well aware of the body of work supporting dual learning-rate models, and have previously also found dual-learning rate models to provide better fits to behavior. However, two points are noteworthy:

- Our results are robust: in six separate instances (three experiments * two phases) we find the dual learning-rate model to be inferior to the winning Kalman-filter type models.
- To our knowledge, most previous research did not compare Kalman-filter and dual learning-rate models. It is therefore unknown to what extent Kalman-filter models provide superior fits in previous studies.

We now dedicate a paragraph in the Discussion to elaborate on the issue of dual learning-rate models vs. the Kalman-filter approach (lines 870-891).

Minor Points

1/ the title is also debatable

Thank you for this comment. We chose the title because it directly relates to the interaction between estimation-uncertainty and the valence of affected options.

2/ I suggest renaming the RELATIVE model as the REFERENCE model, as the concept of outcome relativity also applies to the RANGE model.

see doi.org/10.1016/j.cobeha.2021.06.006

Thank you for this suggestion. While we decided to keep the name of the RELATIVE model as it was in their original paper ¹, we now mention the alternative name in the Methods and added your suggested reference (lines 260-261).

1. Palminteri S, Khamassi M, Joffily M, Coricelli G. Contextual modulation of value signals in reward and punishment learning. *Nature communications* **6**, 8096 (2015).

Reviewer #2 (Remarks to the Author):

This manuscript describes one original experiment and two reanalyses of publicly available datasets to evaluate the hypothesis that how often an option is sampled and an observation observed in the context of an instrumental choice task (i.e. estimation uncertainty) influences choice preferences beyond learning and valuation. The purpose of the original experiment was to setup learning contexts in sampling of choice options was deliberately yoked to learning different rates of reinforcement. In doing so the authors could evaluate how learning related performance and estimation uncertainty influenced choice preferences after learning and across multiple reinforcement schedules during an unexpected choice test in which participants had to choose between options in terms of learned valuations. The main result relies on use of model comparisons to infer that learned avoidance of options associated with punishment is at least in part due to sampling those options less. This is shown in both the original experiment, and demonstrated again by applying the modeling approach to the two published datasets that used a similar learn and test task design. Overall I think this an important point that is being made by the authors in a straightforward manner. I have a few concerns:

1. One criticism is that they do not directly evaluate their claims by testing them in a task where sampling of good and bad options is equalized (e.g. through forced choice trials. Instead their claims rely entirely on demonstrating increased predictability of choices in the test scenario when comparing Kalman filter models that include or exclude estimation uncertainty. More discussion of the potential limitations of the modeling approach is warranted.

Thank you for these comments.

First, we would like to emphasize that our results and claims do not hinge on the modeling approach because we also provide model-agnostic results in the form of correlations between the selection frequency during learning and subsequent decision biases during testing (Fig. 2E-H, 3F-G, 4F-G). These findings are replicated in three experiments. The purpose of the modeling is to add a mechanism to explain the behavioral results, namely learning-induced differences in estimation-uncertainty. The rationale for using the sampling-rate as model-agnostic results are described in Methods (lines 207-216), presented in Fig. 2E-H, 3F-G, 4F-G, and mentioned in the Introduction (lines 71-75). We also highlight the model-agnostic nature of our results in the Results for experiment 1 (lines 572-574, 620-623), experiment 2 (lines 668-670), and experiment 3 (718-722). We now also indicate the difference between the model-agnostic results in the Discussion (lines 788-791).

Second, you are completely right that using forced choice trials to equalize the sampling of good and bad options would effectively remove the impact of sampling uncertainty during the testing phase (presuming that there is no bias in how participants process / attend the different feedbacks). Such a manipulation would completely change the task however, and other decision strategies need to be applied (such as range-adaptation). We address this issue in a new paragraph in the Discussion (lines 844-856).

2. There is no discussion of the recent and relevant work by Anne Collins and colleagues examining the dissociable contributions of reinforcement learning and working memory to choice. There is a direct intersection between sampling and working memory, in that items sampled less often will be more prone to being forgotten and choice relationships more difficult to recall. To that end, and because the post-learning choice tests were unannounced, do the authors see differential effects of estimation uncertainty when choosing between pairs of bad or good options based on the block order in which learning occurred? Specifically, we're more recently learned options more or less susceptible to the influence of estimation uncertainty?

Thank you for this highly relevant insight. We are well aware of the work by Anne Collins, and the task was designed to reduce the impact of block order on decision biases. As stated in the Methods (lines 174-178):

“Note that we only created pairs from objects within the same block to minimize the impact of different delays between the learning and the test phase (e.g. associations learned in the first block may be remembered differently as compared to associations learned in the third block).”

Accordingly, all good vs. good and bad vs. bad comparisons were conducted using objects presented within the same learning block.

However, we do admit to not having considered that the time since learning (i.e. as indicated by learning block) may have affected the expression on different decision strategies. For this reason, we re-analyzed the data (i.e. the ANOVA displayed in Table 7) using learning-block as a covariate. In brief, this updated analysis show no significant interactions with learning block (all p-values > 0.109).

We have updated the manuscript addressing this concern, including relevant references by Anne Collins (Methods: lines 175-178 Results: lines 611-614). We also updated Table 7 with the new results.

3. Considering that learning and estimation uncertainty were yoked in the original experimental design and the evidence that learning was weaker in the probabilistic conditions, do the authors claims hold if the deterministic learning conditions are excluded from the analyses? In these learning situations the subjects would only need to receive deterministic feedback in a limited number of trials before they could infer the value of both options presented (one good and one bad). I'm curious as to whether any self-reports were collected from the subjects indicating they were aware or not of the yoked reinforcement contingencies, or if there was behavioral evidence to indicate condition dependent differences in when they stopped sampling bad options. For example, I doubt the participants were showing matching behavior but in such a scenario estimation uncertainty should have much less influence, correct?

Thank you for these comments.

First, we would like to point out that there were no deterministic learning conditions. $p_{\text{Appetitive}} = 1$ or 0 indicates that the feedback will be respectively sampled deterministically from the pool of appetitive (+1, 0) or aversive (0, -1) feedbacks. However, the feedback is then drawn with a probability of 0.75 from the assigned pool, e.g. positive feedback in the $p_{\text{Appetitive}}=1$ condition will be +1 or 0 with probabilities 0.75 or 0.25. To clarify this uncommon design, Table 1 provides the probability of drawing each feedback type for each object and condition.

Second, unfortunately we did not collect any self-reports regarding strategies or beliefs regarding the experimental setup but will do better in future studies.

Finally, we believe our task was relatively difficult to fully learn, given rather few trials (16), probabilistic learning (probability of getting positive feedback was 0.75), different p_{Approach} between condition, and three possible feedbacks (-1, 0, +1). Indeed, the average behavior in the final trials of all conditions were between 0.7-0.8, which is more indicative of matching, rather than maximizing behavior (which would be indicative of stopping sampling).

In summary, it is difficult to know to what extent a simpler task would yield different results, or to what extent a better understanding of the experimental setup would affect the impact of estimation uncertainty on the learning and the test phases.

Reviewer #3 (Remarks to the Author):

This manuscript reports on the effects of expected value and uncertainty (specifically, estimation uncertainty) on choice behavior in a decision-making task. By using a specific task design – where different pairs of stimuli, learned in separate blocks, have the same value difference but different absolute expected values, and then assessing choices between stimuli from different blocks in a separate test phase – the authors are able to separately assess effects of expected value and unexpected uncertainty on choice, even after learning. Overall, this is a well-done and interesting paper. Most of my comments are targeted towards clarifying and strengthening the findings.

1. The discussion is rather superficial and mostly restates the results, and the end of the introduction is also weak. In particular, the manuscript does not discuss implications of the findings for theories of decision-making. Although I'm sure the word count is partially responsible, bringing in the following areas, if possible, would strengthen the manuscript:

a. Effects of losses on exploration: studies have found asymmetries in exploratory behavior when avoiding losses vs. seeking gains (e.g., Krueger, Wilson, & Cohen, 2017; Lejarraga & Hertwig, 2017). Do the findings in this paper help explain these effects?

Thank you for raising this topic. Unlike most previous studies reporting asymmetric exploration during learning in gain and loss conditions, most of our conditions during learning are intermixed (i.e. feedbacks can be positive, negative, and neutral). Accordingly, while our findings cannot be used to inform previously observed exploration effects between gain and loss conditions, our study provides testable predictions for future studies:

- First, increased exploration during learning leads to more equally distributed sampling of good and bad options. Accordingly, in a post-learning test phase without feedback, conditions with high exploration rates should display reduced biases between decision strategies based on expected values and estimation uncertainty.
- Second, loss conditions have been found to increase exploration during learning.

Accordingly, using a similar setup as in the present study (i.e. learning with partial feedback followed by testing without feedback) we predict:

- For options drawn from gain conditions during learning, good-good / bad-bad decisions should respectively depend more on expected values / estimation uncertainty.
- By contrast, this bias should be reduced for options drawn from loss conditions, because good / bad options are sampled less / more frequently than in gain conditions.

This point is now addressed in the Discussion (lines 892-901).

b. Implications of the Kalman-QU model explaining the data better than other models

Thank you for requesting this clarification. The major implication is that estimation uncertainty needs to be considered when trying to understand human decision making in similar tasks. This point is repeated throughout the manuscript: Abstract (lines 23-28), Introduction (lines 80-83), Discussion (lines 839-843). Another implication is that learning in similar tasks may be better described by a learning rate which changes dynamically across the learning session, rather than presuming constant scaling of prediction errors which is the more common approach. This feature of the Kalman-filter model is specifically addressed in the Discussion (lines 870-891).

2. Related to 1b above, does the addition of the U part of the choice model to the non-Kalman models improve prediction?

Thank you for this very interesting question. We created new versions of all non-Kalman filter models by adding estimates of estimation uncertainty, as obtained from the Kalman-filter approach, to the decision phase of each model. We then performed pairwise comparisons between each old model (X:Q) and its new counterpart (X:QU) for all experimental phases (six instances: three experiments x two phases). In brief, adding estimation uncertainty enhanced model fits to behavior in 29/30 comparisons.

These results have been added to the Results (lines 753-759) and the Discussion (lines 839-841). The analysis is described in detail in Supplementary note 2 and the outcomes of the model comparisons are shown in Supplementary Figure 1.

3. The methods are well done – good model checking and use of model-free and model-based method. I have a few optional suggestions:

a. I was surprised to not see (or perhaps I missed) more interpretation of the parameters from the computational model – in particular, that participants are indeed uncertainty averse during both learning and test phases, but especially so (possibly significantly so?) in test relative to learning phases.

Thank you for this suggestion. We now provide a full analysis of model-fitted parameters, with a particular focus on between-phase comparisons (i.e. learning versus test phase).

In brief, there was no clear consistency between experiments. One study showed large differences in β_Q between learning and test phases, while two others did not. By contrast, two studies found more negative β_U during testing, while the third study did not.

These results are now mentioned in the Results (739-752) and described in detail in Supplementary Note 1, with standardized decision weights displayed in Supplementary Figure 1A,B.

b. In addition, in the results section, when discussing post-learning (test phase)

decisions, it would be helpful to briefly review what learning performance is and its relation to outcome uncertainty. Otherwise the analyses can be confusing to follow.

Thank you for this suggestion. We now provide a description of the relationship between learning performance, sampling-rate, and estimation uncertainty in the Results section (lines 569-572).

4. Minor: the methods state each block has 90 trials, but should this be 80 trials (16 trials per condition x five conditions)?

Thank you – this error has been corrected.

5. For figure 2D, it may be helpful to sort the x-axis comparisons by entropy or otherwise highlight that most comparisons with the largest difference between good and bad are those with two high-entropy choices.

Thank you for this highly relevant suggestion. Please observe that the labels on the x-axis were slightly offset in Fig. 2D. This offset may have suggested that the largest differences between good-good and bad-bad pair were related to differences in entropy.

Inspired by your suggestion, we directly tested the relationship between differences in selection rates during testing and differences in expected values and entropy during learning:

- For expected values, we observed a significant positive relationship for good-good pairs, but not for bad-bad pairs.
- For entropy, we found a significantly positive relationship for bad-bad pairs, but not for good-good pairs.

These discrepancies between good-good and bad-bad pairs further justifies looking into the relationship between learning performance and subsequent test performance.

We corrected the x-axis in Fig. 2D, and now present the results of the correlation analysis in the Results (lines 552-562).

Krueger, P. M., Wilson, R. C., & Cohen, J. D. (2017). Strategies for exploration in the domain of losses. *Judgment and Decision Making*, 12(2), 104-117.

Lejarraga, T., & Hertwig, R. (2017). How the threat of losses makes people explore more than the promise of gains. *Psychonomic bulletin & review*, 24, 708-720.

We sincerely thank the reviewer for their careful and insightful review of our manuscript, which have significantly increased its quality. Below, we address each of their concerns in a point-by-point fashion. The reviewer's comments are displayed in black font, while our responses are in blue font.

Reviewer #1 (Remarks to the Author):

Review for the Revision of Aberg et al. ("Better the Devil..." etc.)

I must express some slight disappointment with the generally low level of engagement and the limited modifications made by the authors in response to my suggestions. Some of my comments were dismissed as a "matter of taste." But I find it difficult to justify presenting a 16-panel figure as an effective and clear way to communicate key results. Similarly, the utility of an exceedance probability figure is questionable (as it does not show actual data points). That being said, I will focus my comments on the authors' responses to my substantive points 1 and 2.

1/ Selection of Experiments in Bavard et al. 2021

While the authors justify their selection of experiments to re-analyze based on their similarity to their own, the selection remains confounded by the omission of experiments where their model would fail. This may raise in other readers concerns about cherry-picking supporting examples from the literature. I recommend that the authors be extra-transparent about this and include analyses of the other experiments from Bavard et al. 2021. This would facilitate a more honest discussion of the points at issue. After all, the authors themselves concede that different cognitive processes explain different outcomes in different tasks, so why not show the full picture?

Thank you for this comment. It is important to emphasize that the two selected experiments were not just based on the 'similarity' to our own, but because they were the only experiments of Bavard et al. (2021) that could support or weaken our hypotheses. To be more precise, the main topics addressed by our study was whether and how sampling biases, naturally arising during partial feedback conditions, affect decision making in subsequent no-learning conditions. Accordingly, because there are no sampling-biases during complete feedback conditions, four of their eight experiments were excluded. In the four remaining experiments, feedback was presented during the test phase in two of them, something which confounds decision biases caused by the initial learning phase with on-going learning and decision strategies during the test phase (e.g. exploration and confirmatory checking behaviors). Accordingly, these two experiments were also excluded.

Therefore, we selected all of the experiments from Bavard et al. (2021) which could provide information regarding our main hypotheses. Cherry-picking here would have been to include only one of those two experiments, while disregarding the other. We feel that the use of the term 'cherry-picking', which carry a strongly negative connotation, was unfairly and unnecessarily used here.

That said, as per your suggestion, we have performed a model-based analysis of behavior in the six other experiments provided by Bavard et al. (2021). Our analysis identified a couple of issues, as described briefly below:

1. To our understanding, Bavard et al. (2021) fit their models across the eight different experiments. This approach makes it impossible to ascertain whether the 'optimal' models differ between experiments. To account for this, we fit the models to each experiment separately (as in our main study).
2. Bavard et al. (2021) fit the parameters to the learning phase and then use these parameters to assess behavior during the test phase (i.e. parameters were actually never fitted to test phase behavior). However, no evidence was provided showing actual similarities between learning phase and test phase model parameters. For example, people explore when feedback is available, a behavior which is usually reflected in smaller decision weights for expected values. It is therefore unlikely that the model parameters for conditions where feedback is presented are the same as in conditions where no feedback is presented (and where no exploration occurs). To overcome this issue, we fit different parameters for the learning phase and for the subsequent test phase (as in our main study).
3. Similarly, besides different parameters, participants may also apply different behavioral strategies, as reflected by different optimal models, in conditions where feedback is available (versus not available). To overcome this issue, we allow for models during the learning phase to differ from the subsequent test phase.

The outcomes of this analysis is now reported in the main text (lines 764-772), described in Supplementary Note 4, with the results being displayed in Supplementary Figure 2.

Particularly noteworthy is that the Kalman:QU model is again the preferred model in the partial feedback conditions, thus supporting the main results. Moreover, while there are consistencies within feedback conditions (i.e. across mixed- and blockwise presentations), there are differences between experiments. We therefore uncover additional insights about behavior using existing datasets. We see this as good scientific practice of using other published datasets to further strengthen the new experiments we performed.

2/ Magnitude Effect

I could have been more precise in my initial review. While it is true that a significant magnitude effect is reported in Bavard et al. 2021, its small size is inconsistent with absolute value encoding and instead requires a range normalization process. Additionally, in a recent study (Anllo et al. 2024) using a similar task, the magnitude effect disappears entirely. My question to the authors is: Does their uncertainty avoidance model account for the very small magnitude effect observed during the learning phase of Bavard et al. 2021? Without resorting to ad hoc modifications (e.g., different temperatures for different pairs of options), I suspect it cannot. If this is the case, the authors may find themselves in the uncomfortable position of needing two different explanations for the learning effects (small magnitude effect) and transfer phases result (preference for C in BC).

Thank you for being more precise. To address your question, we assessed the ability of the model to replicate the magnitude effect by first comparing the difference in average learning performance between each high EV pair and each low EV pair within each of the two tested experiments. For both actual behavior and for model-fitted behavior we observed significantly better performance for high EV pairs in all comparisons (Supplementary Table 1; all p-values < 0.05). Beyond group-level analyses, we also assessed the ability of the model to replicate behavior on an individual level by correlating the difference in model-fitted performance with the difference in actual behavior (i.e. the difference between the learning performance for high versus low EV pairs). Indeed, all correlation coefficients were significantly positive (Supplementary Table 1; the smallest Pearson's $r=0.72$, all p-values<0.001).

In summary, our model reproduces the small magnitude effect in both experiments, on both the group level and on the individual level.

This analysis is reported in the main text (lines 757-763), Supplementary Note 3, as well as in Supplementary Table 1 which shows the results.

3/ Task-Specific Explanations

The authors argue that their model explains key behavior in some tasks (partial feedback) but not others (complete feedback). While this position is tenable, it risks disappointing readers who expect models to offer general and universal explanations. By rejecting range normalization—which works across all tasks—in favor of task-specific explanations, the authors are, in a broader sense, giving up parsimony.

Thank you for this comment. While we would not like to disappoint readers, the main focus of our study is not to provide a general/universal model. Parsimony is indeed a desired feature, yet can be an oversimplified explanation when the assumptions and simplifications fail to capture the underlying complexity of the data and/or identify that different conditions are driven by different underlying factors. Instead, we successfully highlight a significant oversight in the interpretation of the results of numerous previous studies which try to assess learning biases using partial feedback conditions. This is why we conducted our main experiment, and validated those results using two additional relevant experiments (rather than re-analyzing all the experiments of Bavard et al. (2021)).

Moreover, our results, showing a better fit for uncertainty-based models in partial feedback conditions, are actually a challenge to the claim that range-adaptation models provide a general and universal description of behavior across tasks. We show that uncertainty-based explanations provide consistently better descriptions of behavior. While it would be interesting to develop the uncertainty-based models further, we believe the reviewer appreciates this topic to not only be beyond the scope of the present study, but also a significant and full project by itself.

Moreover, this interpretational challenge becomes more significant in light of evidence showing that the key effect (preference for C in BC comparisons) is

stronger in complete feedback experiments, even within-subject (see Palminteri et al. 2015; Nahuel-Salem 2023). These studies show that the same individuals exhibit stronger inversions in complete feedback conditions compared to partial feedback. Are the authors not surprised we require different explanations for two very similar behavioral effects occurring within the same subject at the same time?

We are not surprised that different situations may evoke different behavioral strategies, in fact, this is a very common finding and well known fact in all of behavior. For example, one argument in favor of uncertainty-based models is that it reduces the cognitive load required to solve the task. Specifically, tracking how frequently an object has been sampled is arguably less taxing than estimating the average reward within a context, or dynamically track the lowest and highest possible outcomes within a context, and then apply a re-scaling of each feedback before evaluating prediction errors.

After careful consideration, can't the results mentioned above (Palminteri et al., 2015; Nahuel-Salem 2023), which show stronger inversions in complete (versus partial) feedback conditions, be interpreted as evidence that different strategies are being applied in these conditions? In other words, if the same strategy was applied in partial and complete feedback conditions, wouldn't you expect the same magnitude of inversion effects?

Our results clearly and robustly show that uncertainty-based models explain behavior better in partial feedback conditions, and because these models cannot explain behavior in complete feedback tasks, this current evidence suggest different strategies are likely applied in partial vs. complete feedback conditions. Future studies with carefully designed experiments are needed to disentangle if and when participants apply strategies based on uncertainty-aversion versus range-adaptation.